# A branching model of lineage differentiation underpinning the neurogenic potential of enteric glia

Anna Laddach [1,11] ✉, Song Hui Chng [1,6,11], Reena Lasrado[1,7], Fränze Progatzky [1], Michael Shapiro[1], Alek Erickson[2], Marisol Sampedro Castaneda [3], Artem V. Artemov[4,8], Ana Carina Bon-Frauches[1], Eleni-Maria Amaniti[1,9], Jens Kleinjung[1,10], Stefan Boeing[5], Sila Ultanir [3], Igor Adameyko [2,4] & Vassilis Pachnis [1] ✉

Glial cells have been proposed as a source of neural progenitors, but the mechanisms underpinning the neurogenic potential of adult glia are not known. Using single cell transcriptomic profiling, we show that enteric glial cells represent a cell state attained by autonomic neural crest cells as they transition along a linear differentiation trajectory that allows them to retain neurogenic potential while acquiring mature glial functions. Key neurogenic loci in early enteric nervous system progenitors remain in open chromatin configuration in mature enteric glia, thus facilitating neuronal differentiation under appropriate conditions. Molecular profiling and gene targeting of enteric glial cells in a cell culture model of enteric neurogenesis and a gut injury model demonstrate that neuronal differentiation of glia is driven by transcriptional programs employed in vivo by early progenitors. Our work provides mechanistic insight into the regulatory landscape underpinning the development of intestinal neural circuits and generates a platform for advancing glial cells as therapeutic agents for the treatment of neural deficits.

Neurogenesis and gliogenesis are commonly formalized as alternative and irreversible cell fate decisions made by bipotential progenitors. However, glial cells often share molecular markers with neural stem cells and have the capacity to differentiate into neurons under certain conditions[1]. For example, populations of mature glial cells undergoing constitutive neurogenesis have been identified in specific locations of the peripheral nervous system, such as the carotid body of rodents and the intestine of zebrafish[2,3]. How glial cell lineages acquire mature characteristics while retaining properties of undifferentiated neuronal progenitors, is unclear. The enteric nervous system (ENS) encompasses the gut-intrinsic neuroglial networks that regulate vital gastrointestinal functions, including motility, epithelial secretion and immunity[4,5]. Most enteric neurons and glia originate from a small population of autonomic neural crest cells (ANCCs) which invade the foregut during embryogenesis and, following highly coordinated programs of self-renewal and differentiation, generate integrated

[1]Nervous System Development and Homeostasis Laboratory, the Francis Crick Institute, 1 Midland Road, London NW1 1AT, UK. [2]Department of Physiology and Pharmacology, Karolinska Institutet, Stockholm 17165, Sweden. [3]Kinases and Brain Development Laboratory, the Francis Crick Institute, 1 Midland Road, London NW1 1AT, UK. [4]Department of Neuroimmunology, Center for Brain Research, Medical University of Vienna, Bienna 1090, Austria. [5]Bioinformatics and Biostatistics Science Technology Platform, the Francis Crick Institute, 1 Midland Road, London NW1 1AT, UK. [6]Present address: Experimental Drug Development Centre A*STAR 10 Biopolis Road, Chromos 138670, Singapore. [7]Present address: COMPASS Pathways PLC, Fora, 33 Broadwick St, London W1F 0DQ, UK. [8]Present address: Boehringer Ingelheim RCV, Vienna, Austria. [9]Present address: Sainsbury Wellcome Centre, London, UK. [10]Present address: Sosei Heptares, Steinmetz Building, Granta Park, Great Abington, Cambridge CB21 6DG, UK. [11]These authors contributed equally: Anna Laddach, Song Hui Chng. ✉e-mail: Anna.Laddach@crick.ac.uk; Vassilis.Pachnis@crick.ac.uk

networks of diverse neuronal and glial cell types throughout the gut[6,7]. The critical roles of the ENS in survival and health are highlighted by developmental deficits, such as Hirschsprung's disease (congenital megacolon), which results from a localised but life-threatening absence of neurons and glia from the distal colon[8]. Relatively common conditions, such as intestinal inflammation or aging can also lead to severe gastrointestinal malfunction due to loss of enteric neurons[9,10]. An ideal scenario for restoring ENS function in these conditions would entail the reactivation of developmental programs capable of inducing heterochronic differentiation of resident neural crest-derived cells into appropriate neuronal and glial cell types. Although in mammals enteric neurogenesis is completed shortly after birth, we and other groups have presented evidence that enteric glial cells (EGCs) from postnatal (including adult) rodents can differentiate into neurons in gut injury models and in culture[11–14]. Despite these studies, it is unclear how the differentiation processes of the ENS, unfolding over protracted periods of prenatal and postnatal development, enable mature EGCs to perform their canonical neuroregulatory roles and immune functions in the tissue environment of the adult gut and simultaneously serve as facultative neural progenitors.

Here we characterize at single-cell resolution the gene expression and chromatin accessibility profile of mammalian ENS progenitors and their descendants at key developmental stages and adulthood. These experiments lead us to suggest a previously unknown configuration of ENS lineage differentiation trajectories which proposes that, in contrast to a widely assumed bifurcation model, neurogenic trajectories branch off a progenitor-to-glia differentiation axis. Cells transitioning along this axis progressively lose their initially strong neurogenic bias and adopt gene expression profiles characteristic of canonical glial and immune functions. Nevertheless, by performing molecular profiling of EGCs and their differentiation progeny in a cell culture model of enteric neurogenesis and a gut injury mouse model, we demonstrate that mature EGCs can undergo neuronal differentiation by reactivating neurogenic programs that are employed in vivo by early ENS progenitors. Based on epigenetic profiling of ANCCs and EGCs, we propose that the neurogenic potential of mature enteric glia is encoded by the open chromatin configuration of key neurogenic gene promoters, which enables their efficient reactivation under appropriate conditions. Our model highlights the dynamic character of cell states along the glial differentiation trajectory of the ENS and rationalises the ability of EGCs to perform neuroprotective and immunoregulatory roles while preserving their neurogenic potential. This work sets the foundations for the molecular dissection of developmental processes underpinning the neural control of gastrointestinal function and facilitates the design of strategies to harness mature glial cells for the treatment of congenital or acquired neural deficits.

## Results

### Transcriptional profiling of ENS cells during development

We performed single cell RNA sequencing (scRNA-seq) of tdTomato[+] (tdT[+]) cells from the small intestine of Sox10CreER|tdT reporter mice[13,15] at different developmental stages and adulthood (Fig. 1a and Methods). This transgenic reporter labels undifferentiated ENS progenitors, committed neuronal precursors and EGCs[15,16]. tdT[+] cells from embryonic day (E)13.5, E17.5 and postnatal day (P)1 formed adjoining clusters of undifferentiated progenitors and committed neuronal precursors (Fig. 1b and Supplementary Fig. 1a). However, early (E13.5) tdT[+] cells were clearly separated from their late (E17.5/P1) counterparts, suggesting distinct transcriptional profiles. Indeed, cluster 1 co-expressed progenitor and neuronal markers, revealing a strong neurogenic bias of early ENS progenitors, while clusters 3 and 4 downregulated the neuronal markers and upregulated *S100b* and *Plp1*, indicating that late progenitors are characterized by diminished neurogenic output and

acquisition of glial characteristics (Fig. 1c). This was supported by k-means clustering ($k = 2$) with the progenitor and neuronal gene markers, which showed that the representation of committed neural precursors in the tdT[+] cell population was progressively reduced from E13.5 to P1 (Fig. 1d). Similarly, there was a decrease in the proportion of proliferative cells over time (Supplementary Fig. 1b). tdT[+] cells labelled at E12.5 but sequenced after 3 days of lineage tracing (rather than 24 h) clustered with E17.5 cells and were composed mainly of committed neuronal precursors (Fig. 1d, e), indicating that early and late tdT[+] cells originated from the same population of SOX10[+] ENS progenitors. Finally, tdT[+] cells from P26 and P61 animals were intermixed in two quiescent clusters of glial cells[16] (Fig. 1b), which maintained expression of progenitor gene markers but upregulated the glial markers *S100b* and *Gfap* (Fig. 1c). Consistent with the lack of neurogenesis in the adult mammalian ENS under steady state conditions[12,13], no committed neuronal precursors emerged from the P26/P61 glial clusters (Fig. 1b). These studies demonstrate that during mammalian development a common pool of SOX10[+] ENS progenitors and their descendants transition collectively to a quiescent cell state characterised by reduced neurogenic output and acquisition of glial properties.

### Branching model of cell fate decisions in the ENS

Using lineage and pseudotime inference algorithms[17,18], we next identified a developmental time-aligned progenitor-to-glia ("gliogenic") differentiation trajectory and progenitor-to-neuron ("neurogenic") trajectories, which emerged orthogonally from the gliogenic axis (Fig. 1f and Supplementary Fig. 1c). This previously unknown configuration of ENS differentiation trajectories suggested that the neurogenic axes branch off a gliogenic backbone that maintains a relatively consistent directionality of gene expression change (DGEC) throughout development. We quantified DGEC along the neurogenic and gliogenic trajectories by developing an R package (TrajectoryGeometry; https://bioconductor.org/packages/devel/bioc/html/TrajectoryGeometry.html) which allowed us to project segments of pseudotime paths along an arbitrary number of dimensions of the PCA space onto an n-1 dimensional sphere[19]. The diameter of the circle indicating the mean spherical distance of projection points from a reference centre is a measure of the degree of dispersion of these points in space. A trajectory that maintains course will show a smaller circle relative to a trajectory that changes direction (e.g. one which includes a branch point), which will be represented by a larger circle (Supplementary Fig. 2a). Sampling 1000 paths over the same developmental time (E13.5 to adult) revealed that, although the neurogenic and gliogenic trajectories showed more consistent directionality relative to random trajectories, the gliogenic axis maintained a more consistent DGEC in comparison to the neurogenic trajectory (Fig. 1g, h; Supplementary Fig. 2b, c; Supplementary Data 1–3). Furthermore, the stabilization of the mean distance values for neurogenic trajectory segments commencing at successively later points in pseudotime supported the presence of a branch point, after which the directionality of the trajectory becomes more significant (Fig. 1g, i, j; Supplementary Fig. 2d). We suggest that the branch point corresponds to the commitment of ENS progenitors to neurogenic differentiation. A similar pattern was not evident for the gliogenic trajectory (Supplementary Fig. 2d), which does not change course and therefore shows no obvious point of commitment to glial differentiation. Together, our experimental and computational analyses suggest that during mammalian development ENS progenitors either commit to neurogenic paths or proceed along a linear differentiation pathway giving rise eventually to mature enteric glia. Therefore, ENS progenitors appear to make a single decision, namely to commit to the neurogenic lineage or not, and those progenitors that do not commit to the neurogenic lineage give rise to glial cells by default.

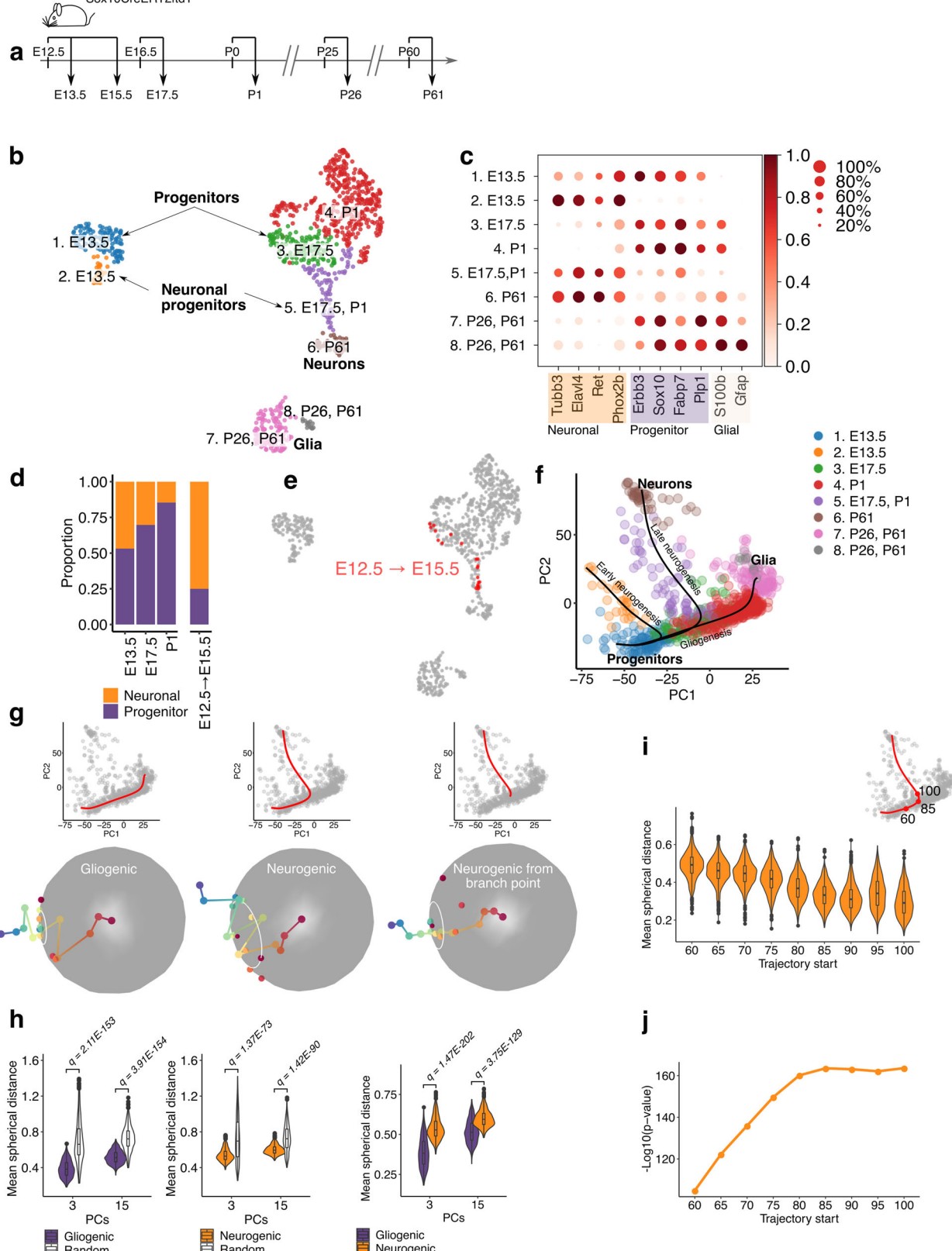

## Gene expression dynamics along the gliogenic trajectory

To uncover the transcriptional basis for the transition of ENS progenitors from a neurogenic to glial character, we used ANTLER (Another Transcriptome Lineage Explorer)[20] (https://juliendelile.github.io/Antler/) to identify sets of genes with co-ordinate patterns of expression across the data (Supplementary Fig. 3; Supplementary

Data 4). E13.5 ENS progenitors were uniquely characterised by co-expression of gene modules (GMs) related to cell cycle regulation (GM1-5) and neurogenesis (GM16-18) (Fig. 2a, b and Supplementary Fig. 3). GM16-18 included genes encoding WNT signalling components, *Phox2b* and *Hmga2* (and its target *Igfbp2*), which are required for enteric neurogenesis[21,22] (Fig. 2a, b, d and Supplementary Fig. 4a).

**Fig. 1 | scRNA-seq and TrajectoryGeometry support a branching model of ENS lineage development. a** Developmental timeline for labelling and isolation of SOX10+ ENS cells. **b** UMAP representation of sequenced cells (904) coloured by cluster. **c** Dot plot representing level of expression of neuronal, progenitor and glial markers in clusters shown in (**b**). The colour scale represents the mean expression level; dot size represents the percentage of cells with non-zero expression within a given cluster. **d** Stacked bar plot showing neuronal and progenitor fractions within the cell populations isolated at the indicated timepoints. **e** The UMAP of panel b showing SOX10+ cells labelled at E12.5 and isolated at E15.5. **f** Slingshot analysis indicating the differentiation trajectories of ENS lineages, depicted on a PCA plot. **g** Individual paths for the gliogenic, neurogenic and post-branching neurogenic trajectories shown on the PCA plot (top) and projected onto a sphere (bottom). The radius of the white circles indicates the mean spherical distance from the centre of the projections. **h** Violin plots indicating the mean spherical distance (radii of the white circles in **g**) for paths sampled from the gliogenic and neurogenic trajectories (purple and orange, respectively) relative to random trajectories (white) and to

each other. Statistics (two-sided Wilcoxon signed-rank for comparison to random trajectories and two-sided Mann–Whitney *U* test for neurogenic/gliogenic comparison) calculated using 1000 paths sampled from each trajectory. **i** Violin plots indicating the mean spherical distance for the neurogenic trajectory starting from successively later points in pseudotime, as the branch point is approached (85 value on the neurogenic trajectory shown in the top right inset). Calculated using three principal components. In (**h**) and (**i**) the box centre represents the median, lower and upper hinges correspond to the first and third quartiles (the 25th and 75th percentiles). The whiskers extend from the hinge to the largest value no further than 1.5 * inter-quartile from the hinge. Points beyond the end of these are plotted individually. **j** Line graph indicating the −log10(*p*-value) for the significance of directionality (two-sided Wilcoxon signed-rank tests) for the neuronal trajectory relative to random trajectories, starting from successively later points in pseudo-time. Calculated using three principal components. Source data are provided as Source Data files.

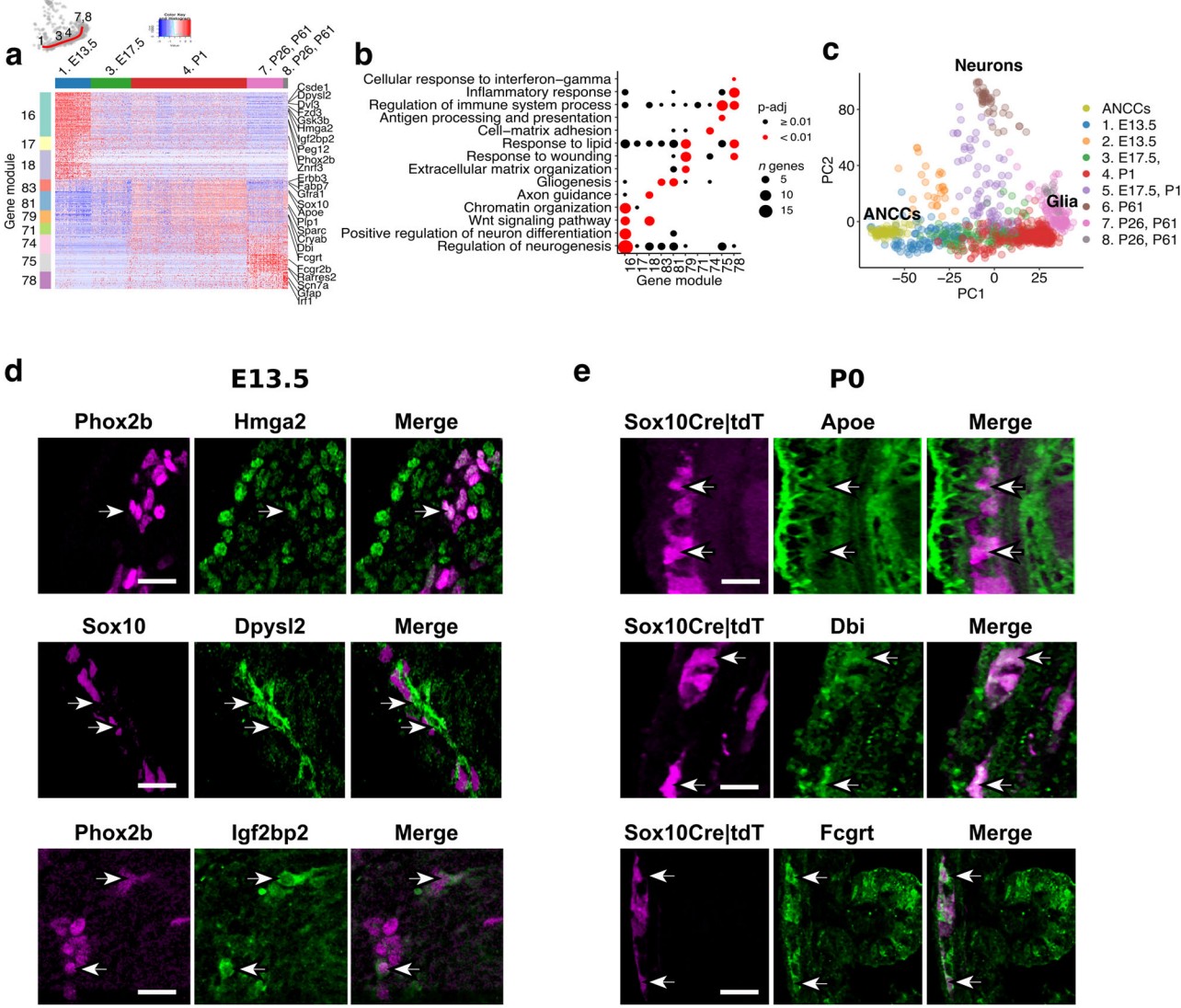

**Fig. 2 | Transcriptional changes along the gliogenic trajectory. a** Heatmap (scaled normalised expression) for selected gene modules associated with transcriptional changes along the gliogenic trajectory (red line on the PCA; top left). Representative genes are indicated on the right. **b** Dot plot showing statistical significance (colour of dot) and size of overlap (size of dot) between selected gene modules and indicated GO terms. Statistics have been calculated using Fisher's one-tailed test and *p* values have been adjusted for multiple comparisons. **c** PCA plot of scRNA-seq datasets from tdT+ ENS cells (904) (Fig. 1b, e) and ANCCs (94). **d, e** Immunostaining for the validation of expression of genes from ANTLER GMs in the ENS of mice at E13.5 (**d**) and P0 (**e**). Arrows point to cells of the ENS lineages. Scale bars: 50 µm. Immunostainings were performed twice independently with similar results. Source data is provided as a Source Data file.

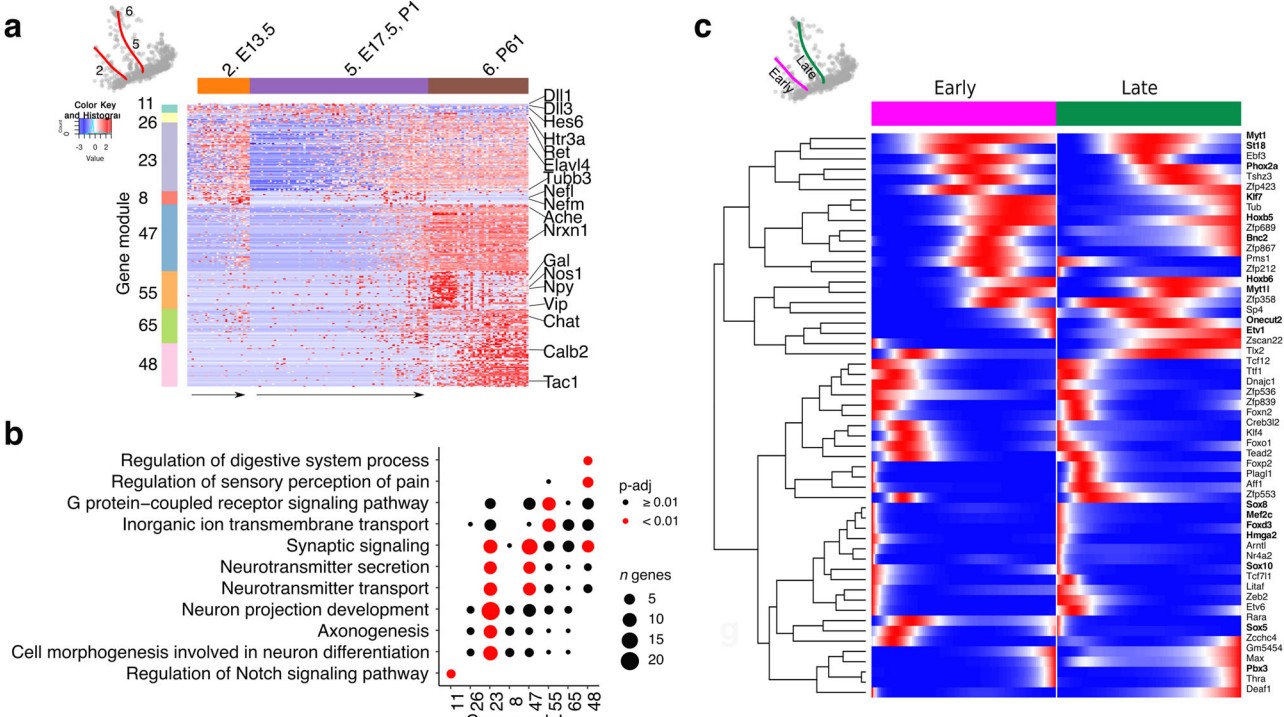

**Fig. 3 | Transcriptomic analysis of early and late neurogenic trajectories of the ENS. a** Heatmap (scaled normalised expression) for selected gene modules associated with early and late post-branch neurogenic trajectories (red lines on the PCA; top left). Representative genes are indicated on the right. **b** Dot plot showing statistical significance (colour of dot) and size of overlap (size of dot) between selected gene modules and indicated GO terms. Statistics have been calculated using Fisher's one-tailed test and *p* values have been adjusted for multiple comparisons. **c** Smoothed expression profiles for genes differentially expressed over pseudotime for early (magenta) or late (green) post-branch neurogenic trajectories. Genes shown in bold overlap with those reported in ref. 41. Source data is provided as a Source Data file.

Interestingly, GM16 also included *Csde1* (an RNA binding protein that inhibits neurogenesis in a cell culture model[23]) and several of its target transcripts (e.g. *Dpysl2*)[24,25] (Fig. 2a, d and Supplementary Fig. 4a; Supplementary Data 5–8). We suggest therefore that the effective neuronal differentiation output of E13.5 progenitors is controlled by the integrated activity of transcriptional programs that drive neurogenesis and posttranscriptional checkpoints that constrain neuronal differentiation, presumably to allow for the controlled expansion of the initially small ENS progenitor pool.

To determine whether the neurogenic bias of early ENS progenitors is an emerging property induced by the tissue microenvironment of the embryonic gut, we performed a combined analysis of our previously reported transcriptome of ANCCs[26] with the current scRNA-seq datasets. ANCCs expressed the neurogenesis-associated modules GM16-18 (Supplementary Fig. 5a) and overlapped with E13.5 progenitors on the principal component analysis plot (Fig. 2c), indicating that they represent the incipient population of the gliogenic trajectory. Interestingly, apart from *Phox2b*, GM16-18 genes were also expressed by diverse neural crest cell populations[26] (Supplementary Fig. 5b). Therefore, the neurogenic bias of E13.5 ENS progenitors is an intrinsic property of the ANCC lineage, which is licensed to undergo neurogenesis by the sequential induction of a small number of transcriptional regulators.

Relative to ANCCs and E13.5 progenitors, E17.5/P1 progenitors downregulated the neurogenic modules GM16-18 and upregulated GMs 83, 81 and 79, which are related to glial development and the communication of cells with their tissue environment (Fig. 2a, b and Supplementary Fig. 3). Upregulated genes included known markers and regulators of gliogenesis (*Sox10, Apoe, Erbb3, Fabp7, Plp1*)[27–31], extracellular matrix organization (*Sparc*)[32], response to wounding (*Cryab*)[33], and lipid metabolism (*Dbi*)[34] (Fig. 2a, b, e and Supplementary Fig. 4b). In agreement with the emerging roles of enteric glia in gut

immunity and host defence[16, 35], GMs 71, 75, 78 (which included genes associated with the regulation of immune function and inflammatory responses, such as *Fcgrt, Scn7a, Ifi27l2a, Rarres2, Serping1, Cxcl1, Irf1*)[36–41] were upregulated specifically in EGCs (Fig. 2a, b, e). Interestingly, several genes in these GMs represent targets of IFN-γ signalling, suggesting a role of this pro-inflammatory cytokine in the induction and maintenance of the immunoregulatory function of enteric glia. This analysis demonstrates that the transcriptional landscape of the gliogenic trajectory is highly dynamic and changes according to the emerging anatomical and functional requirements of the gut: as development proceeds, the initially strong neurogenic character of SOX10+ cells, which ensures the efficient generation of enteric neurons, is replaced by cellular profiles befitting the roles of this lineage in supporting the nascent intestinal neural circuits and maintaining immune homeostasis.

### Shared transcriptional programs during early and late ENS neurogenesis

Next, we examined whether the state of SOX10+ cells along the gliogenic axis impacts on the transcriptional programs driving neuronal commitment and differentiation. We found that ANTLER GMs related to neuronal differentiation (GM 26, 23, 8 and 47) and cardinal specification of enteric neurons to cholinergic (GM55) and nitrergic (GM65) subtypes were expressed by committed neuronal precursors and immature neurons at each time point we analysed and in mature neurons (Fig. 3a, b and Supplementary Fig. 3). Also, 55 transcription factors identified using Monocle's gene test function differentialGeneTest[42,43], including many previously reported to be involved in enteric neurogenesis[44], showed similar expression dynamics along the early and late neurogenic trajectories (Fig. 3c). Among these were two clusters of transcription factors that were restricted to the early stages of pseudotime and included known

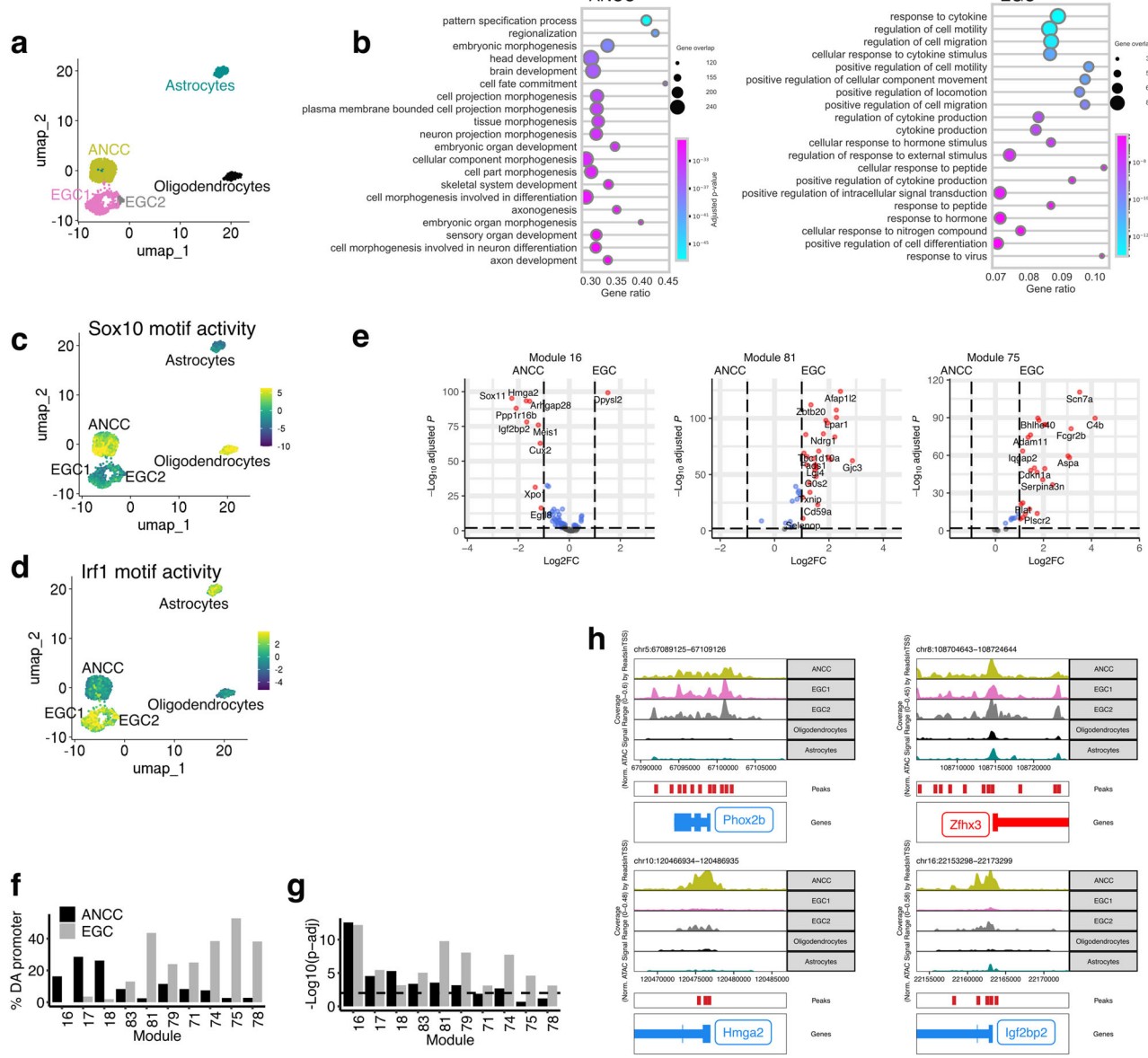

**Fig. 4 | Epigenetic changes along the gliogenic trajectory. a** UMAP representation of scATAC-seq for EGCs (686), ANCCs (964), cortical astrocytes (246) and cortical oligodendrocytes (214). **b** Dot plot showing GO terms overrepresented among genes with differentially accessible peaks (log2FC >1 & padj <0.01) in promoter regions for ANCCs and EGCs. Dot size indicates the overlap for each term, and gene ratio indicates the fraction of genes in each term. **c, d** UMAPs (as in panel **a**) indicating SOX10 (**c**) and IRF1 (**d**) motif activity, calculated using Chromvar. **e** Volcano plots showing mean log₂-transformed fold change (FC; *x* axis) and significance (−Log₁₀(adjusted *P* value)) of differentially accessible (DA) genes from the indicated GMs between ANCCs and EGCs. **f** Bar plot showing the percentage of genes in the indicated GMs (*x* axis) that have at least one DA peak in their promoter region between ANCCs and EGCs. **g** Bar plot showing the statistical significance (hypergeometric test, −Log₁₀(*p*-adj)) of enrichment of genes with at least one peak in their promoter region for the indicated GMs. Dashed line indicates *p*-adj = 0.01. **h** Track plots of ATAC signals for genes maintaining accessibility in EGCs relative to ANCCs (*Phox2b*, *Zfhx3*) and genes with reduced accessibility in EGCs relative to ANCCs (*Hmga2*, *Igfbp2*). Source data are provided as Source Data files.

regulators of enteric neurogenesis (such as SOX10, FOXD3, HMGA2) (Fig. 3c). These analyses suggest that, despite the diminishing neurogenic output of SOX10⁺ cells advancing along the gliogenic axis, they share the potential to activate, albeit with waning efficiency, common regulatory programs that drive the commitment, differentiation and maturation of enteric neurons during development.

## Shared and divergent chromatin accessibility patterns of ANCCs and EGCs

To examine whether epigenetic remodelling is implicated in cell state transitions along the gliogenic trajectory, we compared the chromatin accessibility profiles of ANCCs and EGCs by performing single-cell assay for transposase accessible chromatin using sequencing (scATAC-

seq)⁴⁵. The two cell populations were clearly separated from each other and from astrocytes and oligodendrocytes on the UMAP, indicating distinct chromatin accessibility profiles (Fig. 4a). Genes with promoter regions more accessible in ANCCs were associated with GO terms related to neural development, tissue morphogenesis and cell fate commitment. In contrast, gene promoters more accessible in EGCs were associated with terms related to cytokine responses and cell migration (Fig. 4b). Therefore, in general, the chromatin accessibility and transcriptional profiles of ANNCs and EGCs were concordant. To explore further the regulatory roles of the ANCC and EGC chromatin landscapes on gene expression, we searched for transcription factor motifs enriched in differentially accessible chromatin regions. As expected, among the top hits in ANCCs were motifs for known

transcriptional regulators of neural crest cell and ENS development (TFAP2a, TFAP2c and SOX10)[46, 47] (Fig. 4c and Supplementary Fig. 6a; Supplementary Data 9). In contrast, EGC chromatin peaks were enriched in motifs for FOS and JUN, family members of the generic stress response AP1 transcription factor (Supplementary Fig. 6a; Supplementary Data 10), and interferon regulatory factors (IRFs)[48] (Fig. 4d and Supplementary Fig. 6a), consistent with the role of the IFNγ-EGC signalling axis in maintaining immune homeostasis of the gut[16]. Therefore, chromatin remodelling of SOX10+ cells and differential utilization of transcription factor binding sites along the gliogenic trajectory is associated with the transition of ANCCs from effective neurogenic progenitors to enteric glia, which, in addition to their canonical neuroprotective roles, maintain intestinal tissue homeostasis and modulate immune responses.

To determine whether EGCs preserve features of chromatin organisation characteristic of early ENS progenitors, we compared the chromatin accessibility of genes from the stage-specific GMs of the gliogenic axis between ANCCs and EGCs. As expected, GM16-18 genes were overall more accessible in ANCCs while genes from GM 83, 81, 79, 71, 74, 75 and 78 were more accessible in EGCs (Fig. 4e and Supplementary Fig. 6b). However, when considering the accessibility of individual peaks, it became apparent that in the case of GM16 genes the majority of differentially accessible (DA) chromatin regions mapped to intronic and distal elements and only very few mapped to cognate promoters (Supplementary Fig. 6c). Specifically, the percentage of GM16 genes with promoter regions more accessible in ANCCs or EGCs was 16.3% and 0%, respectively (Fig. 4f). Strikingly, in EGCs 81.5 % of GM16 genes (including *Phox2b* and *Zfhx3*) contained at least one peak in their promoter (Fig. 4h and Supplementary Fig. 6d), significantly more than expected by chance (Fig. 4g; hypergeometric test, *p*-adj = 6.2E−13, see Methods for details). The remaining (18.5%) GM16 genes (which included the chromatin remodelling factor *Hmga2* and its target *Igf2bp2*) lacked promoter peaks in EGCs (Fig. 4h). These findings suggest that the promoters of the majority of the genes that drive the neurogenic output of early ENS progenitors maintain an open chromatin configuration throughout the gliogenic axis, thus enabling EGCs to maintain a memory of their neurogenic past and undergo neuronal differentiation under appropriate conditions. Furthermore, this model raises the possibility that the handful of GM16 genes with reduced promoter accessibility in EGCs are key regulators of neurogenic activity along the gliogenic axis. Interestingly, ANCCs also had more GM83, 81, 79 genes with peaks in their promoters (82.6%, 76.9%, 80%, respectively) than expected by chance (GM83 *p*-adj = 4.17E-4, GM81 *p*-adj = 2.76E-4, GM 79 *p*-adj = 6.67E-4) (Fig. 4g and Supplementary Fig. 6d), suggesting that to some extent the chromatin organisation of these cells anticipates the onset of the gliogenic gene expression profile. In contrast, for GM75 and 78 only EGCs had significantly more genes (77.8%, 70.6%) containing at least one peak in the promoter region than expected by chance (GM75 *p*-adj = 2.45E−5, GM75 p-adj = 7.51E−4) (Fig. 4g and Supplementary Fig. 6d), arguing that cues from the gut tissue environment are necessary for the chromatin remodelling associated with immune gene expression in EGCs.

## Neurogenic cultures of EGCs

Our analyses so far argue that the gliogenic trajectory of the mammalian ENS represents a continuum occupied by dynamic transcriptional and chromatin states that correspond to hierarchies of roles assumed by SOX10+ cells within the tissue environment of the embryonic and postnatal gut. Therefore, we posited that mature mammalian EGCs are capable of returning to upstream positions of the gliogenic axis and reactivating neurogenic transcriptional programs employed by early ENS progenitors. To test this idea, we developed a robust cell culture model of EGC activation and neuronal differentiation (see Methods for details) (Fig. 5a). Within 4 days after plating (days

in vitro-DIV) lineally marked (tdT+) EGCs re-entered the cell cycle and acquired morphology of early ENS progenitors (Fig. 5b). At the end of the culture period (DIV20), we observed interconnected ganglia-like clusters (called hereafter "ganglioids") which were composed of cells expressing neuronal (TUJ1) and glial markers (S100B) (Fig. 5c). Glia-derived neurons also expressed markers of synaptogenesis (SYN1) and mature neuronal subtypes, such as NOS, VIP, CALB and NPY, which were upregulated upon addition of retinoic acid (RA), a signalling molecule required for enteric neuron development in vivo[49] (Fig. 5d–f). In addition to upregulating the expression of neuronal subtype markers, patch clamp analysis demonstrated that RA also promoted the electrophysiological maturation of ganglioid neurons (Fig. 5g, h).

To examine how the gene expression profile of EGC-derived cells was unfolding in ganglioid cultures, we performed RNA-seq of freshly isolated EGCs (DIV0) and their tdT+ descendants from DIV4, DIV11 and DIV20 ganglioids. Bulk RNA-seq showed that relative to DIV0, DIV4 cells downregulated the mature enteric glial gene modules GM75 and GM81 and upregulated the neurogenic module GM16 (Fig. 5i, j), indicating that shortly after plating EGCs lose their glial character and acquire properties associated with early ENS progenitors. The highest upregulated GM16 genes at DIV4 were *Hmga2* and *Igfbp2* (Fig. 5j), which already at this stage had acquired an open chromatin configuration (Fig. 5k; compare to Fig. 4h), suggesting that promoter accessibility of these genes is an early epigenetic change required for neuronal differentiation of enteric glia. To further characterise the process of cellular differentiation in ganglioid cultures, we performed scRNA-seq. UMAP analysis clearly separated DIV0 EGCs (clusters 0 and 1) from the DIV4 cluster 2 (Progenitor-like cells; Fig. 5l and Supplementary Fig. 7a) and confirmed the rapid downregulation after plating of gene modules GM75 and GM81 and several glial markers (such as *S100b* and *Scn7a*), the upregulation of the neurogenic module GM16 and induction of cell proliferation markers, such as *Mki67* (Fig. 5m Supplementary Fig. 7a, b, d, e). A subset of DIV10 cells formed clusters 3 and 4 (Fig. 5l and Supplementary Fig. 7a), which also upregulated *Ret* and *Ascl1* (Supplementary Fig. 7b, c, f), suggesting that they represent cells committed to neuronal differentiation. These cells presumably give rise to DIV20 cells in cluster 5 (Neurons; Fig. 5l and Supplementary Fig. 7a), which expressed gene modules characteristic of in vivo neurogenesis and mature neuronal subtype markers, such as *Nos1* (Supplementary Fig. 7b, e, f, g). Therefore, EGC-derived ganglioid cultures give rise to neurons by following a differentiation trajectory that is analogous to in vivo neurogenesis. The remaining of DIV10 and DIV20 cells formed clusters 6–9 (Fig. 5l and Supplementary Fig. 7a), which expressed EGC markers, such as *S100b* and *Apoe* (Fig. 5m and Supplementary Fig. 7b, f). However, even at DIV20 these glia-like cells expressed relatively low levels (GM81) and failed to reactivate (GM75) gene modules that characterise mature EGCs in vivo (see above and Fig. 5m), arguing that, similar to chromatin accessibility (Fig. 4g), key features of the transcriptional profile of enteric glia (in particular those related to their immune character) depend on signals from the tissue environment of the gut. Nevertheless, a PCA plot revealed that EGCs, progenitor-like and glia-like cells form a backbone axis from which the neurogenic trajectory branches off, in a configuration analogous to that observed in vivo (Fig. 5n).

To further validate ganglioids as a cell culture model of mammalian enteric neurogenesis, we used CRISPR-Cas9-mediated gene editing and pharmacological inhibition in vitro to examine the role of *Ret* and *Ascl1*, key regulators of ENS development. These experiments demonstrated that ablation of *Ret* and *Ascl1* and addition of a RET inhibitor to the culture media, resulted in a significant reduction in the percentage of neurons generated in culture (Fig. 6a–c and Supplementary Fig. 7h–j). In addition, ganglioid cultures established from *Sox10CreERT2;Foxd3fl/fl* mice (see Methods), had a reduced number of

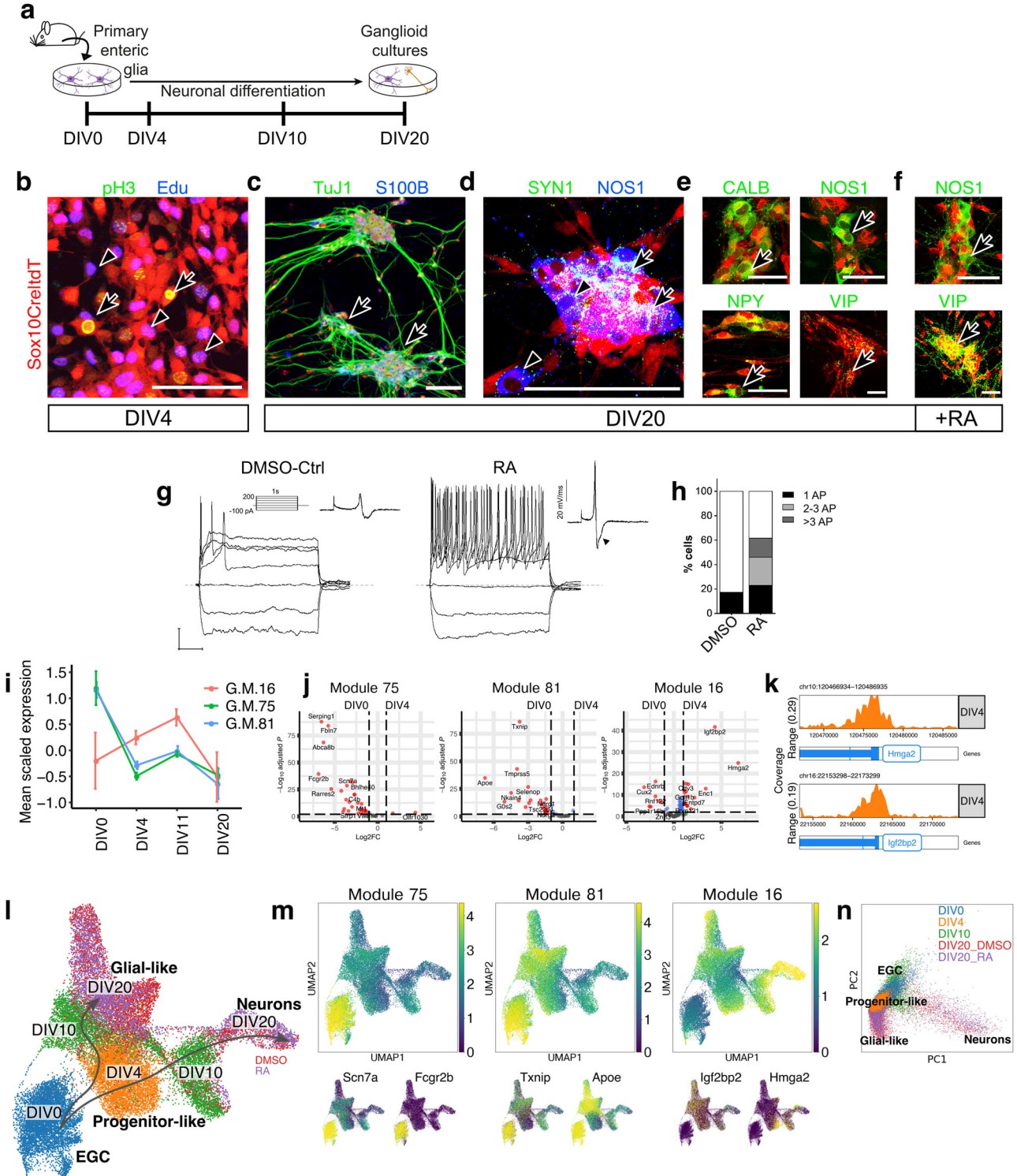

neurons (Fig. 6d, e and Supplementary Fig. 7k, l), consistent with the role of *Foxd3* in maintaining multipotency and neurogenic potential of neural crest cells in vivo[50,51].

Previous studies have indicated that infectious challenges and chemical injury of the gut by dextran sodium sulphate (DSS) or benzalkonium chloride (BAC) of the gut can activate the expression of neuronal markers[13,14]. To examine whether gut injury can induce in EGCs neurogenic programs similar to those expressed by early ENS progenitors in vivo, we performed bulk RNA-seq of enteric glia isolated from mouse guts 48 h after BAC treatment, as described previously[13] (Fig. 6f and Supplementary Fig. 8a–f). We found that BAC-induced

injury activated in EGCs the expression of genes related to "cell cycle", "cell differentiation" and "nervous system development" (Fig. 6g, h). Interestingly, BAC treatment also induced in EGCs a subset of GM16 genes, such as *Hmga2* and *Igfbp2* (Fig. 6h), which are among the top genes upregulated in ganglioid cultures (Fig. 5j). In contrast, RNA-seq of EGCs from mice infected with the helminth *Heligmosomoides polygyrus* showed activation of genes related to cell cycle and IFN-β and IFN-γ response[16]. These experiments argue that neuronal differentiation of EGCs is not a generic response to cell cycle activation but is rather linked to specific signals associated with gut injury. Taken together, our studies demonstrate that under suitable experimental

**Fig. 5 | Characterization of ganglioid cultures. a** Time points of ganglioid culture analysis. **b** At DIV4, cells derived from tdT⁺ EGCs (red) incorporate EdU (blue, arrowhead) and are labelled by pH3 (green/yellow, arrow). Scale bar: 100 μm. **c–e** Immunostaining of DIV20 ganglioids (arrows) with TuJ1 (green) and S100B (blue) (**c**), SYN1 (green, arrows) and NOS1 (blue, arrowheads) (**d**), CALB, NOS1, NPY and VIP (green, arrows) (**e**). Scale bars: 500 μm (**c**), 100 μm (**d**, **e**). **f** Immunostaining of retinoic acid-supplemented DIV20 ganglioid cultures for NOS1 (green, arrows, top) and VIP (green, arrows, bottom). Scale bars: 100 μm. All immunostainings (**b–f**) were performed at least 4 times. **g** Superimposed membrane voltage responses of representative action potential-generating DMSO (control; left) and retinoic acid-treated (right) neurons following 1 s current injections from −100 to +200 pA in 50 pA increments (scale bars: horizontal 200 ms, vertical 20 mV). Corresponding time derivative waveforms of action potentials are shown as inserts. Arrowhead points to the presence of a hump, which is a defining feature of AH-type enteric neurons (control 0/5 = 0%; RA 14/15 = 93.3%). **h** Bar plot showing proportion of cells firing

action potentials in each treatment group: control (untreated and DMSO-ctrl cells) *n* = 29; RA treated: *n* = 26. **i** Mean (±standard deviation) scaled expression of GM16, 75 and 81 in bulk RNA-seq datasets (three biologically independent samples each DIV0, DIV4, DIV10, four biologically independent samples DIV20) generated from ganglioid cultures at the indicated time points. **j** Volcano plots showing mean log₂-transformed fold change (FC; *x* axis) and significance (−Log₁₀(adjusted *P* value)) of differentially expressed genes from the indicated GMs at DIV0 and DIV4 of ganglioid cultures. **k** Track plots showing *Hmga2* and *Igf2bp2* accessibility in bulk ATAC-seq data collected from DIV4. **l** UMAP representation of sequenced cells (27726) from ganglioid cultures colour-coded according to DIV. **m** Mean expression of developmental time-associated GMs on the UMAP shown in (**l**) (top) and expression of representative genes (bottom). **n** PCA representation of scRNA-seq data for ganglioid cultures coloured by Louvain cluster. Source data are provided as Source Data files. Cell images used in panel 5a were from BioRender.com.

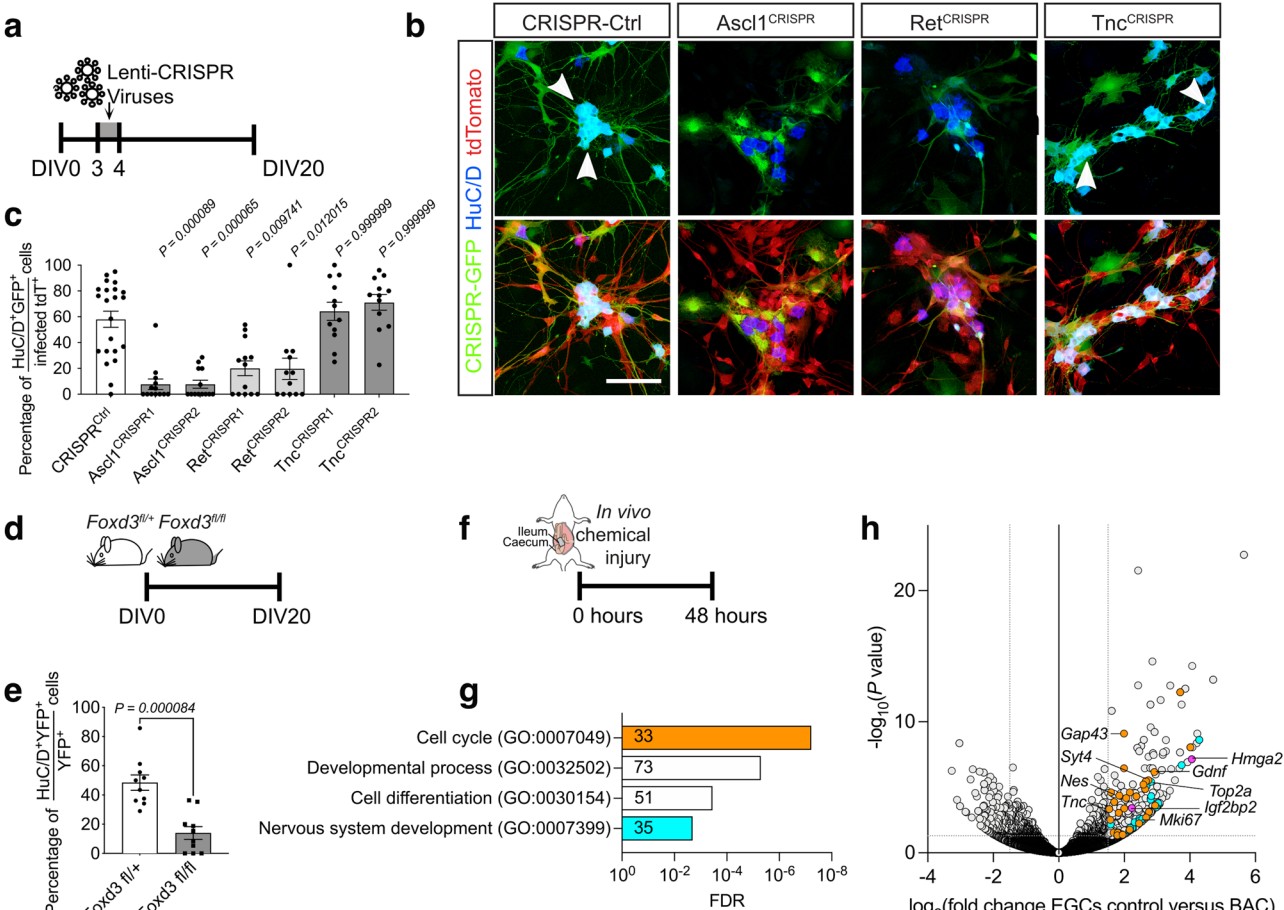

**Fig. 6 | Ganglioid cultures and gut injury recapitulate key features of enteric neurogenesis. a** Experimental strategy for CRISPR editing of ganglioid cultures. **b** Immunostaining of CRISPR-virus infected ganglioid cultures for HuC/D at DIV20. tdTomato (red) indicates cells originating from tdT⁺ EGCs and GFP (green) identifies CRISPR-infected cells (arrows), respectively. The genes targeted by CRISPR are indicated at the top. **c** Quantification of neurons following CRISPR editing of ganglioid cultures at DIV20. Data are mean ± s.e.m. (*n* = 22 (CRISPR^CTRL), *n* = 13 (Ascl1^CRISPR1+2, Ret^CRISPR1), *n* = 12 (Ret^CRISPR2, Tnc^CRISPR1+2)) fields of view per group). Kruskal–Wallis test with Dunn's multiple comparisons test. **d** Experimental strategy for the generation of ganglioid cultures established from *Sox10CreER^T2;Foxd3^fl/+* and *Sox10CreER^T2;Foxd3^fl/fl* mice. **e** Quantification of neurons in cultures from

*Sox10CreER^T2;Foxd3^fl/+* and *Sox10CreER^T2;Foxd3^fl/fl* mice at DIV20. Data are mean ± s.e.m. (*n* = 10, fields of view per group, pooled from two independent experiments). Unpaired student's *t*-test. **f** Experimental strategy for the collection of bulk RNA-seq data after BAC treatment. **g** Bar plot showing GO terms overrepresented amongst genes upregulated after BAC treatment. The number of genes upregulated per GO term are indicated in the bars. **h** Volcano plots showing mean log₂-transformed fold change (FC; *x* axis) and significance (−Log₁₀(adjusted *P* value)) of differentially expressed genes after BAC treatment. Cell-cycle associated genes coloured yellow, nervous system development associated genes coloured cyan and GM16 genes coloured magenta. Source data are provided as Source Data files.

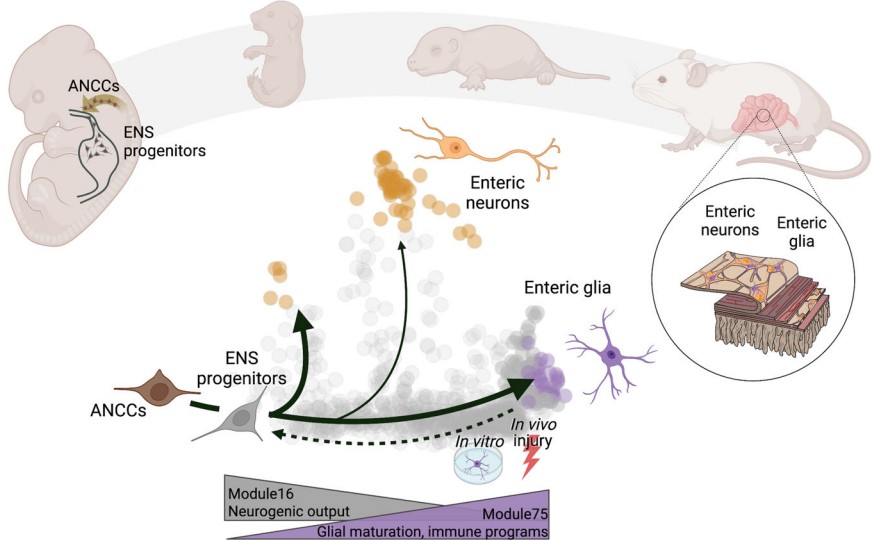

**Fig. 7 | Schematic showing how a branching model of lineage decisions in the ENS underpins the neurogenic potential of adult glia.** ANCCs become ENS progenitors upon invasion of the foregut. A default differentiation trajectory, that maintains a relatively continuous directionality of gene expression change, gives rise to mature enteric glia. Neurogenic trajectories branch off from this default trajectory during embryonic and early postnatal time points. As cells transit along the default ENS progenitor-glia axis neurogenic output diminishes until it ceases in adult animals at homeostasis. Accordingly, transcriptional modules that underlie active neurogenic activity (e.g. GM 16) are downregulated, whereas those related to glial maturation and immune function (e.g. GM75) are upregulated. In response to changes in the environment (in vivo injury, culture conditions) glial cells can transit to upstream positions of their developmental axis and reactivate neurogenic transcriptional programmes. Figure created with BioRender.com.

conditions, mammalian EGCs are capable of activating neurogenic programs that are expressed by early ENS progenitors in vivo.

## Discussion

In this study we have examined how the differentiation trajectories of enteric neurons and glia emerge during mammalian development. Our analysis supports a model in which neurogenic differentiation paths branch off from a linear gliogenic trajectory that traces the founder SOX10[+] ANCC population that invades the gut as it loses progressively its pro-neurogenic and proliferative biases and acquires features of quiescent mature enteric glia (Fig. 7). The direct anchoring of post-mitotic neurogenic trajectories on the gliogenic axis and the positive association between neurogenic output and proliferation, ensures that the overall size and balance of the neuronal and glial cell populations in the gut are regulated effectively by the cell cycle dynamics of SOX10[+] progenitors. Our model also provides an explanation for the long-standing observation that the majority of mutations implicated in Hirschsprung's disease, a relatively common neurodevelopmental abnormality of the ENS, have been identified in genes critical for the proliferation of ENS progenitors and lead to the concomitant loss of enteric neurons and glia[52].

In contrast to the egress of neurogenic trajectories, which is associated with a major shift in the directionality of change in gene expression, cells advancing along the gliogenic axis maintain a core transcriptional program established in ANCCs and early ENS progenitors, whilst they progressively acquire a gene expression profile that allows them to adjust to the shifting anatomical and functional requirements of the host gut. Thus, as their initial neurogenic tasks are completed (driven by a series of neurogenic GMs), ENS progenitors activate genes that allow them to support the nascent neural circuits (gliogenic and glia-associated GMs) and participate in tissue homeostasis and host defence (immune GMs). However, mature EGCs are capable of returning to developmentally upstream positions of the gliogenic axis and re-activating neurogenic programs. This occurs in certain contexts, such as ganglioid cultures of mammalian EGCs or following injury of the mouse gut, whereas EGCs in the adult zebrafish are capable of undergoing constitutive

neurogenesis at steady state[3, 13]. Our findings suggest that the potential of EGCs to return to a neurogenic state depends on the continued expression of central regulators of enteric neurogenesis (*Sox10*), and the open chromatin configuration of key neurogenic genes (such as *Phox2b*). Nevertheless, it remains unclear why mammalian EGCs, contrary to their teleost counterparts, fail to undergo in vivo neurogenesis at steady state or do so inefficiently following gut injury. Previous studies have suggested that although inflammation inhibits neurogenesis in mammals, in zebrafish it functions as a positive regulator of neuronal regeneration in the central nervous system[53]. Furthermore, recently it has been documented that the inflammatory priming of fibroblasts inhibits skin regeneration in a mammalian injury model[54]. These reports, together with the efficient generation of enteric neurons in the immune cell-free environment of EGC-based ganglioid cultures, suggest that the inflammatory ground state of mammalian enteric glia driven by pro-inflammatory cytokines, such as IFN-γ, impedes their ability to activate at steady state their intrinsic neurogenic potential. Further experiments will be necessary to determine the effects of immune signalling on the ability of EGCs to activate their neurogenic potential in vivo.

Emerging evidence suggests that EGCs are a molecularly heterogeneous cell population, both at the transcriptional and chromatin organization level[55,56]. Our current studies do not address the question whether all enteric glia have the potential to undergo neurogenesis. Interestingly, a recent report published while our manuscript was under review, suggested that a subset of EGCs enriched in the myenteric ganglia of adult mice have a gene expression profile and chromatin organization that is consistent with neurogenic potential[55]. Our previous in vivo clonal analysis has demonstrated that individual ENS progenitors are capable of generating all morphological subtypes of enteric glia that occupy distinct locations in the gut wall[15], suggesting that the cellular microenvironment in which descendants of ENS progenitors settle and differentiate is at least partly responsible for the molecular heterogeneity of EGCs. Characterisation of the cell-intrinsic mechanisms and environmental signals that allow all or subsets of EGCs to preserve and reactivate their neurogenic potential could have

far reaching implications for the development of therapies that aim to restore neuronal function compromised by injury or disease in the peripheral and central nervous system.

## Methods

### Animals

All animal procedures were carried out at the Francis Crick Institute in accordance with the regulatory standards of the UK Home Office (ASPA 1986) and the ARRIVE guidelines and approved by the local Animal Welfare and Ethics Review Body (AWERB). Mice were housed and bred under specific pathogen-free conditions (SPF) in individually ventilated cages under a 12 h light–dark cycle at ambient temperature (19 °C–21 °C) and humidity (45–55%). Standard food and water were provided ad libitum. Both male and female animals were used for the experiments. The *Sox10CreER^{T2}* transgenic mice refer to two sublines, SER26 (MGI: 5301107) and SER93 (MGI: 5910373) that have been reported previously[13,15,16]. Generation of the *Foxd3^{fl}* allele (MGI: 3790794) and the *Wnt1Cre2* transgene have been described previously[51,57]. The Cre-dependent reporters used are: *Rosa26-tdTomato* (MGI: 3809524)[58] and *Rosa26-nuclearGFP* (MGI: 5443817)[59]. Sox10CreER| tdT and Sox10CreER|YFP indicate *Sox10CreER^{T2}(SER93);Rosa26-tdTomato* and *Sox10CreER^{T2}(SER93);Rosa26-YFP* mice, respectively. Mice of the desired developmental stage were generated by setting up timed pregnancies. The plug day was designated as E0.5 and date of birth as P0.

### Benzalkonium chloride (BAC) treatment

*Sox10CreERT2(SER93);Rosa26-tdT* (Sox10Cre|tdT) mice were injected with two doses of tamoxifen at 100 µg/g of body weight one week prior to BAC treatment. BAC treatment of the small intestine was performed essentially as described by Laranjeira et al.[13]. Briefly, following administration of analgesics (buprenorphine-0.1 mg/kg; meloxicam-2mg/kg) and general anaesthesia with isoflurane, the abdomen was sterilized with chlorhexidine and a 1.5- to 2-cm ventral midline incision was made to expose the distal ileum. A small piece of filter paper (3 mm × 15 mm) soaked in 0.1% BAC (Sigma-Aldrich) in 0.9% saline solution was laid on the surface of the distal ileum (~5 mm proximal to the ileocecal junction) for 5 min. The treated area was thoroughly rinsed with 5 ml saline, the exposed ileum was placed back into the abdomen, and the midline incision was closed using standard surgical procedures. The animals were analysed 48 h after BAC treatment.

### Generation of ganglioid cultures

Neurogenic EGC cultures were established from Sox10Cre|tdT mice (24 h after a single dose of tamoxifen injection at 100 µg/g of body weight) and *Sox10CreERT2;Foxd3^{fl/fl};Rosa26-eYFP* mice (48 h after two doses of tamoxifen injection at 100 µg/g of body weight). Longitudinal muscle-myenteric preparations were washed thoroughly in DMEM/F12 media containing fetal bovine serum (FBS, 2%, Sigma-Aldrich, F7524), penicillin-streptomycin (100 U/ml, Thermo Fisher Scientific) and 12.5 mM HEPES (12.5 mM, Thermo Fisher Scientific). Post-rinsing, tissues were minced using fine scissors and incubated in a shaker at 37°C for 30 min with Type 1 collagenase (0.5 mg/ml, Sigma Aldrich, SCR103) and DNaseI (0.5 mg/ml, Sigma-Aldrich, DN25) in the buffer described above. Next, trituration of the sample using an unpolished glass pasteur pipette was carried out. Triturated samples were then washed twice with DMEM/F12 media containing 10% FBS with penicillin-streptomycin and plated onto two 60 mm (Corning) dishes coated with fibronectin (20 µg/ml; Sigma-Aldrich, F1141). Seeded cells were cultured overnight in DMEM/F12 media containing 10% FBS and penicillin-streptomycin. Next day, the culture medium was changed to serum-free media that contained DMEM/F12 supplemented with N2 (1%, Thermo Fisher Scientific, 17502048), G5 (1%, Thermo Fisher Scientific, 17503012) and NGF-7S (50 ng/ml, Thermo Fisher Scientific, 13290-010). To allow robust proliferation of glial cells, half of the

medium was replaced with fresh medium on day three. On day four, cells were trypsinized using TrypLE express (Thermo Fisher Scientific, 12604013) and filtered using a 40 µm cell strainer to obtain a single cell suspension. Cells were counted (Countess II; Thermo Fisher Scientific) and then seeded at ~30,000 cells/cm² onto fibronectin coated 60 mm dishes in proliferative medium. The passaged cells were collected using Accutase (Thermo Fisher Scientific, 00-4555-56) on day six and seeded onto Poly-D-Lysine/Laminin/Fibronectin coated 12 mm glass coverslips (Corning, #354087) at ~42,000 cells/cm² in 24-well plates. Upon passaging, cells were maintained in media that constitutes both the proliferative and neuronal differentiating medium which comprises of DMEM/F12, N2 supplement (1%), GDNF (10 ng/ml, Peprotech, 450-44), BDNF (10 ng/ml, R&D systems, 248-BD), NT3 (10 ng/ml, R&D systems, 267-N3), NGF-7S (10 ng/ml), cAMP (0.5 mM, Sigma-Aldrich, D0260), Ascorbic Acid (0.2 mM, Sigma-Aldrich, A4034) and penicillin-streptomycin at a ratio of 1:2. On day eight, the culture medium was completely replaced with neuronal differentiating media. Subsequently, every two days, half of the medium was exchanged with fresh media. To label proliferating cells, cultures were exposed to EdU (10 µM) for an hour before fixation. Cells were consistently maintained at 5% CO₂.

For CRISPR editing, cultures were infected at DIV3 with lentiviruses carrying the Cas9 enzyme and sgRNA sequences to target indicated genes (Fig. 6a–c). For RET signalling inhibition, indicated concentrations of GSK3179106, a selective RET kinase inhibitor[60], or DMSO (control) were introduced on DIV8 and gradually reduced every two days from DIV12 onwards as shown in Supplementary Fig. 7h.

### CRISPR/Cas9-sgRNA design and lentivirus production

Lentivirus vectors were generated using HEK293T cells, which were originally supplied by the European Collection of Authenticated Cell Cultures (ECACC, Cat No 12022001). The specific aliquot of cells used for our experiments was provided by the Cell Services STP of the Francis Crick Institute and was negative for mycoplasma. HEK293T cells were allowed to reach confluence in 10 cm dishes with media containing DMEM with high glucose, GlutaMAX Supplement, pyruvate (Thermo Fisher Scientific, 10569010), FBS (10%) and penicillin-streptomycin. Next, cells were transfected with either a control pL-CRISPR.EFS.GFP (15 µg, Addgene, 57818)[61] or pL-CRISPR.EFS.GFP with cloned sgRNA sequences targeting *Ascl1* (CRISPR1-F 5′-CACCGCAACGAGCGCGAGCGCAACC-3′, CRISPR1-R 5′-AAACGGTTGCGCTCGCGCTCGTTGC-3′, CRISPR2-F 5′-CACCGGAG-CATGTCCCCAACGGCG-3′, CRISPR2-R 5′- AAACCGCCGTTGGGGA-CATGCTCC-3′), *Ret* (CRISPR1-F 5′- CACCGAAGCGACGTCCGG CGCCGCA-3′, CRISPR1-R 5′- AAACTGCGGCGCCGGACGTCGCTTC-3′, CRISPR2-F 5′- CACCGTCTATGGCGTCTACCGTACA-3′, CRISPR2-R 5′- AAACTGTACGGTAGACGCCATAGAC-3′) and *Tnc* (CRISPR1-F 5′- CACCGTGTGACGATGGGTTCACAG-3′, CRISPR1-R 5′- AAACCTGT-GAACCCATCGTCACAC-3′, CRISPR2-F 5′- CACCGCCCGCCGATTGT-CACCACCG-3′, CRISPR2-R 5′- AAACCGGTGGTGACAATCGGCGGG C-3′).

The constructs were provided with a packaging vector (Pax2, 12.5 µg) and the envelope protein VSV-G (3.3 µg, Addgene, #8454) in Opti-MEM medium to which the transfection reagent Polyethylenimine (1 mg/ml, Generon) was added at a ratio of 1:3 of the DNA. Guide RNA sequences were designed using Benchling [Biology Software] (2019; retrieved from https://benchling.com) and cloned based on a publicly available lentiviral CRISPR toolbox protocol[62]. Eight hours post-transfection, the supernatant was discarded and replaced with fresh medium. This was followed by collecting ~10 ml of the filtered supernatant 48 h later. The lentivirus was concentrated with the use of the PEG-it virus precipitation solution (2.5 ml, System Biosciences, LV810A-1) for 2–3 days at 4 °C. Subsequently, precipitated viral particles were centrifuged at 2300 × *g* for 30 min at 4 °C, resuspended in

proliferative media (250 µl) and stored at −80 °C as 50 µl single-use aliquots for future use.

## Real time quantitative PCR of ganglioid cultures

Total RNA was isolated from FACS-enriched tdT+ cells of DIV20 using Trizol LS reagent (Thermo Fisher Scientific, 10296010) and the PureLink RNA Micro Kit (Thermo Fisher Scientific, 12183016) as per manufacturer's instructions. Complementary DNA (cDNA) was generated by reverse transcription using the High-Capacity cDNA Reverse Transcription Kit (Thermo Fisher Scientific, 4368814). RT-qPCR was performed with the synthesized cDNA using Taqman fast universal 2X PCR Master Mix (Thermo Fisher Scientific, 4352042) and Taqman probes (Thermo Fisher Scientific) on a 7500 Fast Real-Time PCR system (Applied Biosystems). The Taqman probes used in this study are the following: *Actb* (Mm02619580_g1), *Acta2* (Mm00725412_s1), *Foxd3* (Mm02384867_s1), *Ngfr* (Mm00446296_m1), *Sox10* (Mm00569909_m1), *S100b* (Mm00485897_m1). $C_t$ values obtained were normalized to β-actin and relative changes in expression were calculated using $\Delta\Delta C_t$ analysis.

## Bulk RNA sequencing of EGCs (BAC treatment and control)

Tunica muscularis tissue from control and BAC-treated mice was incubated with 1 mg/ml collagenase IV (Worthington Biochemicals, CLS-4) in Hanks Balanced Salt Solution (HBSS, Thermo Fisher Scientific, 14170112) at 37 °C for 12 min, followed by wash in ice-cold HBSS and subsequent incubation with papain (10 U, Worthington Biochemicals, PAP2) for 5 min at 37 °C. The digested samples were then washed and resuspended in L-15 medium without phenol red (Thermo Fisher, 21083027) containing penicillin–streptomycin (1%, Thermo Fisher Scientific, 15140122), BSA (1 mg/ml, Sigma-Aldrich, A9418), HEPES pH 7.4 (10 mM, Thermo Fisher Scientific, 15630106), Biowhittaker water (10%, Lonza, BE17-724F) and DNase I (400U, Grade II; Roche, 10104159001), filtered through a 70 µm and 40 µm strainer and subjected to FACS using DAPI (1 µg/ml) for live/dead cell discrimination. 100,033 EGCs from 4 BAC treated mice and 190,856 EGCs from 4 control mice were obtained. RNA from FACS-sorted EGCs was isolated using the PureLink RNA Micro Kit (Invitrogen, 12183016) according to the manufacturer's instructions. Total RNA samples were converted to cDNA libraries using the Ovation RNA-Seq System V2 (NuGEN Technologies, 7102-A01) according to the manufacturer's recomendations. After the amplification procedure, the Qubit dsDNA HS Assay kit (Thermo Fisher Scientific, Invitrogen, Q32854) was used to measure the cDNA concentration. SPIA cDNA (sample volume, 15 µl) was sheared to a size of 200 bp by sonication (Covaris, S220/E220 Focused-Ultrasonicator). The Ovation Ultralow System V2 1-16 (NuGEN, 0344NB-A01) was used to produce libraries from fragmented double-stranded cDNA. The fragment ends were then repaired before ligation of sequencing adaptors, followed by PCR amplification and library purification. The quality and the molarity of the libraries were checked using the Agilent TapeStation, (HS D1000 Reagent, 5067-5585; HS D1000 Screentape, 5067-5584). The final libraries were pooled and sequenced on the HiSeq 4000 system.

## Bulk RNA sequencing of ganglioid cultures

Cultured cells were detached from the plates on DIV 4, 11 and 20 and tdT+ cells were enriched by FACS and collected in tubes containing Trizol LS Reagent (Thermo Fisher Scientific, 10296010). Total RNA from 60,000 sorted cells/sample was isolated via the precipitation method in combination with spin-columns using PureLink RNA Micro Kit (Thermo Fisher Scientific, 12183016) according to the manufacturer's instructions. cDNA was generated using the Ovation RNA-Seq System V2 (NuGen Technologies, 7102-A01) and quantified using the dsDNA Qubit HS Assay kit (Thermo Fisher Scientific, Q32854). This was followed by acoustic shearing of the cDNA to generate 200 bp fragments using Covaris E220 (Covaris). The Ovation Ultralow System V2 1-16 (NuGEN, 0344NB-A01) was used to produce libraries from fragmented double-stranded cDNA, followed by PCR amplification and library purification. The quality and quantity of the final libraries was assessed with TapeStation D1000 Assay (Agilent Technologies). The final libraries were pooled and sequenced on the HiSeq4000 (Illumina) to generate 75 bp single-end reads.

## Bulk ATAC sequencing of ganglioid cultures

Crude nuclei isolated using IGEPAL solution from 50,000 FACS sorted tdTomato cells at indicated times were subjected to tagmentation using the Illumina Nextera Tn5 transposase following published protocol with some modifications[63]. Transposition was carried out at 37 °C for an hour instead of the recommended 30 min for all samples processed. DNA fragments were purified using Qiagen MinElute PCR kit and subjected to PCR amplifications using the NEBNext High-Fidelity 2× PCR Master Mix (New England Biolabs) and appropriate barcoded primers as described[64]. Amplified fragments were further purified using AMPure XP Beads (Beckman Coulter) according to manufacturer's recommendation to remove unamplified primers before sequencing. The resulting ATAC libraries were quantified using the GloMax multi-detection system (Promega) and size distribution assayed using a TapeStation D1000 ScreenTape (Agilent). The libraries were then pooled (4 nM) for sequencing and loaded onto the HiSeq 4000 (Illumina) for paired end 100 bp reads sequencing.

## Single cell RNA sequencing

For the transcriptomic analysis described in Fig. 1, tdT+ cells were isolated at indicated stages from the small intestine of *Sox10CreER^{T2};Rosa26-tdTomato* (Sox10Cre|tdT) mice. For all pre- and perinatal stages (E13.5, E17.5, P1) tdT+ cells were isolated from the entire small intestine (from duodenum to terminal ileum). For the P26 and P61 stages, tdT+ cells were isolated from the muscularis externa of the small intestine (duodenum to terminal ileum). Cells were harvested from equal numbers of male and female mice. For the isolation of E17.5, P1 and P26 cells, animals were injected with tamoxifen 20-24 h earlier, while E15.5 cells were isolated from animals injected with tamoxifen at E12.5. The scRNA-seq datasets for E13.5 and P61 cells have been described previously[15,16]. Expression of tdT was induced by intraperitoneal administration of tamoxifen to pregnant females at E12.5 and E16.5 and mice at P25 and P60 at 100 µg/g of body weight. For P1 cells, tamoxifen was injected intradermally at P0. Tissue was incubated with Collagenase I (1 mg/ml, Sigma, C0130) for 30-45 min and enzyme activity was terminated by rinsing cells with ice cold PBS. The cell suspension was centrifuged and filtered through 40 µm nylon cell strainers (Falcon, 352340). tdT+ cells were isolated by flow cytometry (BD Biosciences Aria Fusion sorter equipped with BDFacsDIVA V8 software) with a 100 µm nozzle and collected in 2% FBS in OptiMEM without phenol red (Thermo Fisher Scientific, 11058021). Dead cells were excluded by labelling with the cell viability dye Zombie Aqua (2 ng/ml, BioLegend, 423101) that was added to the single cell suspension before sorting. For the P0 stage, we also sorted the cells directly into 96-well plates to increase the yield. The protocols for the isolation of tdT+ cells from the small intestine of E13.5 embryos and P61 animals have been reported previously[15,16]. RNA sequencing of isolated tdT+ cells was carried out as described previously[15,16]. The Fluidigm C1 automated microfluidic system was used to capture individual cells into 5–10 µm and 10–20 µm chips. The SMARTer Ultralow RNA kit (Clontech, 634833) was used to reverse transcribe poly(A) RNA and amplify cDNA. ERCC spike-ins (Life Technologies, 4456740) were added at a dilution of 1:80,000. cDNA concentration of each cell sample was quantified using the Promega Quantifluor dsDNA on Glomax system and quality of random samples was checked on high-sensitivity DNA chip on the Agilent Bioanalyser 2100. Amplified cDNA (0.125-0.375 ng) was used for generating sequencing libraries with the

Illumina Nextera XT DNA kit (Illumina, FC-131-1096). All libraries were sequenced as paired-end 75 using the Illumina HiSeq4000 system.

For the transcriptomic analysis of adult EGCs (shown in Fig. 5 and Supplementary Fig. 7), nGFP+ cells were isolated from tunica muscularis preparations from 8-week-old *Sox10CreER^T2^(SER93);Rosa26-nuclearGFP* mice that had been injected with tamoxifen (2 doses of tamoxifen at 100 µg/g of body weight 10 days prior to isolation). Tissue was isolated as described above for isolation of glial cells from BAC-treated animals. For scRNA-seq of these cells we used the 10X genomics platform. Approximately 10,000 tdT+ cells post-FACS (from EGC cultures) and 10,000 nGFP+ cells post-FACS (freshly isolated EGCs) were loaded into a 10x Chromium Chip B (10×3′v3) and partitioned into gel bead-in-emulsions (GEMs) using the 10x Chromium Controller. cDNA was generated following manufacturer's protocol and quantified using the TapeStation D5000 ScreenTape (Agilent). Libraries were prepared according to the manufacturer's guidelines and quality was monitored using both TapeStation (Agilent) and High Sensitivity dsDNA Qubit (Thermo Fisher Scientific) systems. Final libraries were pooled and sequenced using the HiSeq4000 (Illumina).

### Single cell ATAC sequencing

For scATAC-seq of adult EGCs, nGFP+ cells were isolated from tunica muscularis preparations from 8-week-old *Sox10CreER^T2^(SER93);Rosa26-nuclearGFP* mice that had been injected with tamoxifen (2 doses of tamoxifen at 100 µg/g of body weight 10 days prior isolation)[16]. For ATAC sequencing of autonomic neural crest cells, tdT+ cells were isolated from *Wnt1Cre;Rosa26-tdTomato* embryos at E9.5 and E10.5. Cells were dissociated using mechanical and enzymatic disintegration as previously reported[65]. Briefly, heads were separated from trunks by slicing the embryos with a razor blade posteriorly to the otic vesicle, tissue was diced with a razor blade and triturated using a P-1000 pipette and incubated for roughly 15 min with Trypsin/EDTA at 37 °C. The enzymatic digestion was halted using ice-cold 1% FBS in HBSS. Cells were pelleted and filtered using a 40 µm cell strainer. Dissociated cell suspension was sorted Tomato-positive from negative cells using a FACSAria III flow cytometer. The sorted cells were stored in RMPI/DMSO buffer in −80 °C before nuclear isolation.

Nuclei were isolated and scATAC-seq libraries were prepared according to the Chromium Single Cell ATAC Reagent Kits User Guide (10x Genomics; 10xGenomics.com CG000496 Rev A). Briefly, after counting, nuclei concentrations were adjusted to the desired capture number 10,000 based on the number of available nuclei and the desired multiplet rate (according to the 10X protocol). To minimize potential multiplets, we typically aimed to capture <6000 nuclei per channel. Next, 5 µl of the resulting resuspended nuclei were combined with ATAC Buffer and ATAC Enzyme (10x Genomics PN-1000390) to form a transposition mix, which was then incubated for 60 min at 37 °C. A master mix composed of Barcoding Reagent, Reducing Agent B and Barcoding Enzyme (10X PN-1000390) was then added to the same tube as transposed nuclei. The resulting solution was loaded onto a Chromium Chip H(10x Genomics; PN-1000161). Vortexed Chromium Single Cell ATAC Gel Beads and Partitioning Oil were also loaded onto the same Chromium Chip H before attaching a 10x Gasket and placed into a Chromium Single Cell Controller instrument (10X genomics). Resulting single-cell GEMs were collected at the completion of the run (~17 min) and linear amplification was performed on AB cycler: 72 °C for 5 min, 98 °C for 30 s, cycled 14×: 98 °C for 10 s, 59 °C for 30 s and 72 °C for 1 min. Emulsions were coalesced using the Recovery Agent (10x Genomics; 220016), then subjected to Dynabeads (2000048) and SPRIselect reagent (Beckman Coulter; B23318) bead clean-ups. Indexed sequencing libraries were constructed by combining the barcoded linear amplification product with a sample index PCR mix comprising SI-PCR Primer B, Amp Mix and Chromium i7 Sample Index Plate N, Set A (10x Genomics; 3000262). Amplification was performed on an AB cycler: 98 °C for 45 s, for 12 cycles: 98 °C for 20 s, 67 °C for 30 s, 72 °C for 20 s, with a final extension of 72 °C for 1 min. The sequencing libraries were subjected to a final bead clean-up SPRI-select reagent and quantified by Qubit and TapeStation HS1000. Sequencing libraries from ANCCs and adult EGCs were loaded on an Illumina NovaSeq6000 and HiSeq4000 sequencer, respectively, with 2 × 50 paired-end kits using the following read length: 51 bp read 1 N, 8 bp i7 index, 24 bp i5 index (trimmed later on) and 51 bp read 2 N using a loading molarity of 280 pM (Standard loading) or 180 pM (Xp loading).

### Sample preparation and Immunostaining experiments

Embryos (E9.5–E13.5) and embryonic guts (>E16.5) were fixed overnight in ice-cold 4% paraformaldehyde (PFA) and processed further for cryostat sectioning (10 µM) or/and whole mount immunostainings. Briefly, samples were subjected to antigen retrieval using sodium citrate (10 mM) at pH 6.0, permeabilized with 0.3% Triton X-100 phosphate buffer (PBT), blocked with 10% donkey serum in PBT for an hour, incubated overnight with primary antibodies at 4 °C, rinsed at room temperature and further incubated with secondary antibodies for two hours at room temperature. After final washes, the samples were either processed for EdU labelling or mounted for confocal imaging with Vectashield Mounting medium (Vector Labs, H-1000 /H-1200). EdU labelling was done with Click-iT EdU Alexa Fluor 647 kit and the manufacturer's instructions were followed (Thermo Fisher Scientific, #C10340).

In vitro cells cultured on cover slips and/or exposed to EdU (10 µM) for one hour were fixed with 4% PFA at room temperature for 20 min. Fixed cells were rinsed with phosphate buffer, followed by antibody and EdU staining protocols as described above without the use of the antigen retrieval step. Stained coverslips were mounted onto glass slides using ProLong Diamond Antifade mountant (Thermo Fisher Scientific, P36961).

For immunostaining of tunica muscularis tissue from control and BAC-treated adult mice, the distal ileum was harvested and a 1-ml pipette was inserted into the gut lumen to fully extend the smooth muscle, as previously described[66]. Briefly, the smooth muscle/myenteric plexus layer was separated, from submucosal plexus/epithelium layer after a small incision, stretched and fixed in 4% paraformaldehyde for 2 h at 4 °C. The fixed gut tissues were then rinsed in PBS before each was processed separately for staining.

The following antibodies were used: GFP (Rat, Nacalai Tesque, #04404-84, 1:500), GFP (Chick, Abcam, #ab13970, 1:400), Ki67 (Mouse, BD Biosciences #550609, 1:200); TuJ1 (Mouse, Biolegend, #801202, 1:1000), SOX10 (Rabbit, ProteinTech, #10422-1-AP, 1:200), S100 (Rabbit, DAKO, #z0311, 1:500), nNOS1 (Goat, Abcam, #ab1376, 1:300), HuC/D (Mouse, Thermo Fisher Scientific, #A-21271, 1:400), PHOX2B (Goat, R&D systems, #AF4940, 1:250), pH3 (Rabbit, Millipore, #06-570, 1:500), CALB1 (Rabbit, Chemicon, #AB1778, 1:500), VIP (Rabbit, Immunostar, #20077, 1:400), NPY (Rabbit, Biogenesis, #6730-0004, 1:400), SYN1 (Rabbit, Abcam, #ab64581, 1:400), CSDE1 (Rabbit, Abcam, #ab201688, 1:200), GSK3β (Rabbit, Abcam, #ab32391, 1:200), PEG12 (Goat, RayBiotech, #NP_038816.1, 1:200), DPYSL2 (Rabbit, Proteintech, #14686, 1:300), DBI (Rabbit, Aviva Systems Biology, ARP33135_P050, 1:200), APOE (Mouse, Abcam, #ab1906, 1:200), β-CATENIN (Mouse, BD Transduction Labs, #610154, 1:300), HMGA2 (Rabbit, Abcam, #ab97276, 1:300), IGF2BP2 (Rabbit, Abcam, #ab124930, 1:300), SPARC (Goat, R&D Systems, #AF942, 1:400), CRYAB (Mouse, Abcam, #ab13496, 1:250), FCGRT (Rabbit, Thermo Fisher Scientific, #PA5-42871, 1:300). Secondary antibodies were used at a dilution of 1:500 in the block buffer and included donkey anti-rat/rabbit/mouse/goat/chicken conjugated with AlexaFluors 405, 488, 568 or 647 as required (Jackson ImmunoResearch or Thermo Fisher Scientific).

## Microscopy, image analysis and quantifications

Stained sections and/or whole mount guts were imaged with X40 (1.75 NA) and/or X20 (1.0 NA) objectives with the upright Leica single/multiphoton TCS-SP5 confocal microscope, supported by the LAS AF software. Images of cell culture experiments were acquired using the upright Olympus Confocal Laser Scanning microscope FV3000 supported by the FV31S-SW software (Olympus) using standard excitation and emission lasers for visualizing various fluorophores (405, 488, 568, 647). Live-cell imaging was performed with the inverted Leica DMI 4000B microscope, fitted with Leica EL6000 light source and supported by the Micromanager (2.0) software[67]. In addition, the Hamamatsu ORCA-spark digital CMOS camera was used to visualise the reporter (tdT) with a standard filter.

Images were processed using Fiji/ ImageJ (Wayne Rasband, NIH) and/or Adobe Photoshop 2020 (Adobe systems).

In cell culture experiments, the field of view always included areas where enteric 'ganglioids' were visualised for quantification of cells.

## Patch clamp recordings of neurons in EGC cultures

Whole-cell patch clamp recordings were obtained at room temperature from the soma of visually identified glial-derived enteric neurons at DIV25-33. Cells were continuously perfused at 1.5 mL/min with extracellular solution containing 140 mM NaCl, 4 mM KCl, 2 mM $CaCl_2$, mM $MgCl_2$, 11 mM glucose, 10 mM HEPES at pH 7.4. Patch electrodes ($5 \pm 1$ MΩ) were pulled from borosilicate glass and filled with the intracellular solution comprised of 135 mM K-gluconate, 10 mM KCl, 10 mM Na-phosphocreatine, 2 mM $MgATP_2$, 0.3 mM $Na_3GTP$, 10 mM HEPES at pH 7.25. Tetrodotoxin citrate (Abcam, ab120055) was added in some experiments to determine the nature of voltage-gated $Na^+$ channels underlying cell activity. Selected tomato-positive 'neuronal' cells were always part of a 'ganglioid' and identified with a larger cell size and diffuse $tdT^+$ signal. Patch-clamped neurons were identified with the use of the Alexa Fluor 488 (20 μM) dye that was included in the pipetted solution. Signals were recorded with a Multi-clamp 700B amplifier and Digidata 4400a driven by PClamp 10.3 software (Molecular Devices). Data were sampled and digitized at 10 kHz. Passive properties were measured using a 100 ms, −5 mV step from a holding potential of −60 mV. Active neuronal properties were probed in current clamp configuration using a series of 1s-long step current injections from −100 to 400 pA in 50 pA increments every 15 s, whilst injecting sufficient current to maintain membrane potential at −60 mV. Data were analysed using Clampfit 10.7, Origin 2018b. Action potential properties were measured using the first action potential generated with a 100 pA current injection. Hyperpolarization-activated voltage sag was calculated as the difference between the peak hyperpolarization due to −100 pA current injection and the voltage at the end of the 1s step. After hyperpolarization (or AHP) amplitude was measured as the peak hyperpolarization following depolarizing current injections steps.

## Analysis of fluidigm scRNA-seq from small intestine SOX10$^+$ cells

**Alignment and quality control.** Reads were aligned to the Ensembl GRCm38 genome using Tophat2[68]. Transcript counts, including counts for spliced and unspliced reads were obtained using the program Velocyto[69].

Cells expressing more than 2000 genes and less than 10% mitochondrial genes were retained for further analysis. Contaminating epithelial cells ($n = 9$) were filtered from the data based on Cdh1 counts (>= 10 counts). Likewise, mesenchymal cells ($n = 39$) were filtered based on the expression of the marker Meis2 (>= 10 counts) and macrophages ($n = 6$) were filtered using expression of the marker Cd14 (>= 10 counts). After filtering 904 cells were retained, 143 cells (E13.5), 20 cells (E12.5 → E15.5), 129 cells (E17.5), 432 cells (P1), 11 cells (P26) and 169 cells (P61).

Genes detected in fewer than 3 cells or with total counts <10 were omitted from further analysis (23,299 genes were retained). Counts were normalised to counts per million (CPM) and log transformed (after the addition of a pseudocount of 1). Cell cycle scoring and assignment was performed in SCANPY[70] using cell cycle genes from Kowalczyk et al, 2015[71].

Batch effects were visualised using dimensionality reduction techniques (UMAP). This showed good mixing between batches for each time point, thus demonstrating that batch effects did not obscure the biology (Supplementary Fig. 1a).

**Calculation of gene modules (GMs).** Normalised gene dispersions were calculated using SCANPY[70]. Gene module analysis was performed using the package Antler (https://github.com/juliendelile/Antler)[20]. All genes with a normalised dispersion > 0 were used as input to the algorithm; this was done to remove genes with a lower normalised dispersion than expected, which are likely to contribute noise to the analysis. The correlation and consistency cut-offs were set to 0.3 as these cut-offs resulted in compact and homogeneous groupings as revealed by a heatmap (Supplementary Fig. 3). The minimum number of genes a gene should be correlated with and the minimum number of cells with a "positive" binarised level per gene module were both set to three. The resulting modules appear biologically meaningful with a number of modules containing genes with similar known functions (e.g. cell cycle markers, glial markers, neuronal markers). In addition, markers associated with excitatory and inhibitory neurons segregate to different modules. Functional analysis (GO biological process) was performed for each module using g:Profiler[72], accessed programmatically using the gprofiler python module.

**Dimensionality reduction and clustering.** The 2954 genes identified as taking part in gene modules were used as input for PCA analysis in SCANPY[70]. The rationale behind the selection of these genes was that these are most likely to have biological relevance in the developmental process; a similar approach was taken by Delile et al.[20]. The first 15 PCs were selected for dimensionality reduction and clustering using the Louvain algorithm (resolution 1.5). The resulting clustering was manually curated and those clusters with similar transcriptional profiles that separated purely based on the expression of cell-cycle associated markers were merged.

To obtain an estimate of proportions of uncommitted cells and cells undergoing neurogenesis at each time point, kmeans clustering ($k = 2$) was performed over the embryonic and P1 data on the normalised expression of 4 neurogenic (Tubb3, Elavl4, Ret, Phox2b) and 4 progenitor (Erbb3, Sox10, Fabp7, Plp1) markers. This resulted in a cluster with higher expression of neurogenic markers that included committed neuronal precursors, and one with higher expression of progenitor markers that included both gliogenic precursors and uncommitted progenitors.

**Identification of Csde1 targets in gene modules.** iCLIP-seq data, which identified direct RNA targets of human CSDE1 in melanoma cells were obtained from Wurth et al.[25]. In addition, we obtained RNA-seq data on genes differentially expressed in hESCs after CSDE1 knockdown from Ju Lee et al.[23]. To map human gene names from these datasets to murine gene names, we used mappings from the NCBI HomoloGene database (ncbi.nlm.nih.gov/homologene). To test whether gene modules were enriched in RNA targets of Csde1 we used an upper tail hypergeometric test, with arguments to the R *phyper* function as follows:

$x$ = the number of Csde1 RNA targets in the gene module

$m$ = the number of Csde1 RNA targets in our whole dataset

$n$ = the total number of genes in our dataset that are not Csde1 RNA targets and can be mapped to human homologues

$k$ = the number of genes in the gene module which can be mapped to human homologues

To test whether gene modules are biased towards genes which are up- or down-regulated upon Csde1 knockdown, we used a Fisher exact test with the contingency table presented in Supplementary Data 11.

To investigate whether gene modules were enriched in transcripts with Csde1 binding motifs we used a script published previously[24]. This scans transcripts for the consensus Csde1 binding sites [A/G] 5AAGUA[A/G] and [A/G]7AAC[A/G]2 determined by SELEX[73]. Genes names from our data set were converted to RefSeq transcript IDs using the R biomaRt package. We considered a gene to contain a Csde1 binding site motif if any of its mapped transcripts contained such a binding motif. To assess whether modules were enriched in Csde1 binding motifs we used permutation testing.

**Pseudotime inference and velocity calculation.** Pseudotime was inferred using Slingshot[18], setting the starting point to the "eEP" cluster. The difference between the G2M phase and S phase was regressed out of the data prior to pseudotime analysis, to maintain differences between cycling cells and non-cycling cells but to regress out differences between cycling cells. Genes from the identified gene modules were again used as input to the PCA and the top 15 PCs used as features for calculation of pseudotime. Clusters which had been defined in "Dimensionality and clustering" were used as input to Slingshot, with the exception that the small EGC2 cluster was merged with the EGC1 cluster.

Smoothed profiles for the expression of transcription factors over pseudotime for neurogenic trajectories from branch points onwards were obtained by fitting spline curves (Monocle 2.12.0, genSmooth-Curves function with 3 degrees of freedom)[42,43]. At this stage any transcription factors expressed in <10 cells within the trajectory were filtered out. Differentially expressed genes over pseudotime were identified using Monocle's differentialGeneTest function with gene level distributions modelled as negative binomial distributions with fixed variance (option expressionFamily=negbinomial.size()). Only embryonic and P1 time points (stages with active neurogenesis) were used for this analysis.

Stochastic RNA velocity was calculated using the programme scVelo[17]. For inference of dynamics, velocity was calculated using genes identified as taking part in gene modules. These were filtered for genes with at least 20 counts and 10 unspliced counts. 1222 of these were identified to have significant velocity and were used for further calculations.

**Analysis of trajectory geometry.** We asked whether changes in gene expression over pseudotime determine a well-defined direction, consistent throughout the course of a trajectory, or whether there are points where cells change course. To answer this question, we analysed the geometry of each trajectory, taking an approach similar to that described by Shapiro et al.[19]. Specifically, we compared the geometries of the trajectories from "eEP" to "AN" and "eEP" to "EGC". Our rationale here being that both trajectories occupy the same amount of real time, with the initial cells having been sampled at embryonic day 13.5 and the final cells having been sampled from adult animals.

Each trajectory can be represented by $n$ points along pseudotime, $\vec{p_1}, \vec{p_2} \ldots \vec{p_n}$ where each point is represented by coordinates along $d$ principal components, where $d$ is the number of dimensions considered. Unit vectors, representing the difference between the starting point $\vec{p_1}$ and each other point in the trajectory can then be calculated:

$$\vec{s_i} = \frac{\vec{p_i} - \vec{p_1}}{\| \vec{p_i} - \vec{p_1} \|} \tag{1}$$

This results in $n - 1$ unit vectors $\vec{s_i}$ for each trajectory. If these are calculated over three dimensions (i.e. using the first three principal components), the resulting unit vectors can be visualised as points on a sphere. For a trajectory with no change in direction (i.e. all points in

the trajectory fall on a straight line) each unit vector $\vec{s_i}$ will be identical, and will project onto a single point on the sphere. In contrast, unit vectors calculated from a random trajectory, will disperse over the surface of the sphere. Thus the compactness of the area covered by a trajectory, projected onto the sphere, can be taken as a measure of how much the direction of a trajectory deviates throughout its course. Although easiest to visualise in three dimensions, this approach can be extended to an arbitrary number of dimensions.

To obtain points $\vec{p_i} \ldots \vec{p_n}$ representative of each trajectory, we employed a sampling strategy. First, we normalised pseudotime for each trajectory to range from 0 to 100. We then sampled a single cell from each sliding window of 10 pseudotime units. For each trajectory we repeated the sampling procedure 1000 times.

To assess the compactness of the area covered by the unit vectors $\vec{s_1} \ldots \overrightarrow{s_{n-1}}$ calculated for a trajectory, we found the unit vector $\vec{c}$, which minimized the mean spherical distance:

$$msd = \frac{1}{n-1} \sum_{i=1}^{n-1} \cos^{-1}(\vec{c} \cdot \vec{s_i}) \tag{2}$$

Minimisation was carried out using the "optim" function in R, using $\frac{\sum_{1}^{n-1} \vec{s_i}}{\| \sum_{1}^{n-1} \vec{s_i} \|}$ as a starting point for minimisation, and 1000 minimisation steps. The value for $\vec{c}$ which minimises $msd$ represents the direction of motion of the trajectory. We used the value $msd$ for that $\vec{c}$ as a summary statistic to represent the compactness of the area covered by each trajectory.

We compared each empirical trajectory derived from the single cell transcriptomics data to a randomised trajectory. Here, the randomised trajectory is based on the empirical trajectory, however coordinates for each principal component are permuted (i.e. permutations are performed within each column of the matrix where rows represent cells and columns represent PCs).

Projections and summary statistics were calculated for randomised paths as for the empirical trajectories. The summary statistics for empirical (1000 sampled trajectories) and randomised (1000 randomised trajectories) data were then compared using Wilcoxon signed-rank tests. The summary statistics for two empirical trajectories (1000 sampled trajectories each) were also compared to each other using Mann–Whitney U tests.

A representative centre $\vec{c}$, which represents the direction of a developmental trajectory, was found by minimising $msd$ for all centres calculated for the empirical trajectories. Genes were scored based on their association with this direction by calculating $F \cdot \vec{c}$ where $F$ is the matrix of factor loadings for the 15 PCs used as input to Slingshot.

## Analysis of bulk RNA-seq from ganglioid cultures
'Trim Galore!' utility version 0.4.2 (https://www.bioinformatics.babraham.ac.uk/projects/trim_galore/) was used to remove sequencing adaptors and to quality trim individual reads with the q-parameter set to 20. The sequencing reads were then aligned to the mouse genome and transcriptome (Ensembl GRCm38 release-86) using RSEM version 1.3.0[74] in conjunction with the STAR aligner version 2.5.2[75]. Sequencing quality of individual samples was assessed using FASTQC version 0.11.5 (https://www.bioinformatics.babraham.ac.uk/projects/fastqc/) and RNA-SeQC version 1.1.8[76]. Differential gene expression was determined using the R Bioconductor package DESeq2 version 1.14.1[77].

## Analysis of bulk ATAC-seq from ganglioid cultures
Reads from ATAC-Seq fastq files were aligned to the Ensembl GRCm38 genome, peaks mapped and quantified using the nextflow atacseq pipeline version 1.0dev (https://nf-co.re/atacseq). Track plots were created in ArchR[78].

## Analysis of 10X RNA-seq from ganglioid cultures

Reads were aligned to the Ensembl GRCm38 using 10x Genomics Cell Ranger 3.0.2. Data was analysed in SCANPY. Cells with >1000 features and <10% mitochondrial counts were retained for downstream analysis. Predicted doublets were filtered using Scrublet[79]. Contaminating neurons ($n = 35$) were filtered using the expression of the marker Elavl4 (>0 counts) and contaminating macrophages ($n = 99$) were filtered using expression of the marker Cd14 (>0 counts) from DIV0 data. Genes detected in fewer than 3 cells or with total counts <10 were omitted from further analysis (23,178 genes were retained). Counts were normalised to counts per million (CPM) and log transformed (after the addition of a pseudocount of 1). Cell cycle scoring and assignment was performed in SCANPY[70] using cell cycle genes[71]. Highly variable genes were detected in SCANPY and the top 4397 used as input to PCA. The first 20 PCs were selected for dimensionality reduction and clustering using the Louvain algorithm (resolution 0.6). Marker genes for each cluster were detected using a Wilcoxon test, comparing each cluster to the union of all other clusters. Gene scores were calculated for selected Antler modules using SCANPY's score_genes() function, and were visualised in SCANPY.

## Analysis of bulk RNA-seq of EGCs from BAC-treated mice

Trim Galore!' utility version 0.4.2 (https://www.bioinformatics.babraham.ac.uk/projects/trim_galore/) was used to remove sequencing adaptors and to quality trim individual reads with the q-parameter set to 20. The sequencing reads were then aligned to the mouse genome and transcriptome (Ensembl GRCm38 release-86) using RSEM version 1.3.0[74] in conjunction with the STAR aligner version 2.5.2a[75]. Sequencing quality of individual samples was assessed using FASTQC version 1.11.5 (https://www.bioinformatics.babraham.ac.uk/projects/fastqc/) and RNA-SeQC version 1.1.8[76]. Differential gene expression was determined using the R Bioconductor package DESeq2 version 1.24.0[77]. Gene ontology analysis was performed on genes with a log2-transformed fold change (BAC treatment vs control) > 1.5 and $P$adj < 0.05 via the Gene Ontology resource interface (http://geneontology.org/)[80–82].

## Analysis of fluidigm scRNA-seq data from ANCCs

Single-cell sequencing data (fastq files) were obtained from GEO (SRA) under accession GSE129114 for mouse E9.5 *Wnt1^{Cre}/R26R^{Tomato}* trunk/vagal neural crest cells. Autonomic cells were selected based on lineage annotations we reported earlier[26]. As for the small intestine developmental data, reads were aligned to the Ensembl GRCm38 genome using Tophat2[68]. Transcript counts, including counts for spliced and unspliced reads were obtained using the program Velocyto[69]. Counts were normalised to counts per million (CPM) and log transformed (after the addition of a pseudocount of 1). ANCC data was merged with the small intestine developmental data and the 2954 genes identified during the analysis of small intestine data were again selected for dimensionality reduction using PCA. The resulting PCA plot shows cells localise in a time-dependent manner along PC1, with ANCCs localising at lower and overlapping values to E13.5 cells along this PC, reflecting a developmental progression. All ANCC batches localise together, and partially intermingle with cells from the closest time point (E13.5), indicating minimal batch effects. Differential expression between ANCCs and EGCs (union of P61 EGC1 and EGC2 cells), and between ANCCs and eEPs was performed in Seurat V3 using the MAST package[83] and the formula -cell_type + nFeature_RNA.

## Analysis of fluidigm scRNA-seq from neural crest cells

Single-cell sequencing data (raw counts) were obtained from GEO under accession GSE129114 for mouse E9.5 *Wnt1^{Cre}/R26R^{Tomato}* trunk/vagal neural crest cells. Counts were normalised to counts per million (CPM) and log transformed (after the addition of a pseudocount of 1)

using Seurat V4[84]. Cells were labelled with lineage annotations we reported previously[26].

## Analysis of scATAC-seq data from EGCs, ANCCs, astrocytes and oligodendrocytes

Reads for EGCs and neural crest cells (E9.5) were aligned to the mm10 genome using 10x Genomics Cell Ranger ATAC 1.2.0. Fragment files (output from 10x Genomics Cell Ranger ATAC 1.2.0) for the fresh cortex from adult mouse brain (P50) data were obtained from 10x genomics (https://www.10xgenomics.com/resources/datasets?menu%5Bproducts.name%5D=Single%20Cell%20ATAC&page=1&configure%5Bfacets%5D%5B0%5D=chemistryVersionAndThroughput&configure%5Bfacets%5D%5B1%5D=pipeline.version&configure%5BhitsPerPage%5D=500).

Downstream analysis was performed using ArchR[78]. Initially each dataset was investigated separately. Cells predicted to be doublets; cells with <= 10000 fragments; and cells with a TSS enrichment <4 were filtered from further analysis from each dataset. Iterative Latent Semantic Indexing (LSI) was performed and the first 30 dimensions used as input to the UMAP algorithm.

Cell types for the cortical and neural crest data were determined by label transfer using the Seurat FindTransferAnchors() function[84] through the ArchR interface[78]. For the cortical dataset we downloaded annotated adult mouse cortical scRNA-seq data from the Allen Institute for Brain Science from https://www.dropbox.com/s/kqsy9tvsklbu7c4/allen_brain.rds in the form of a Seurat object. Single cell neural crest data was processed as in the section "Analysis of neural crest data". EGC1 and EGC2 clusters in the scATAC EGC data were similarly inferred using our C1 fuidigm scRNA-seq EGC data.

For further analysis, cells labelled as oligodendrocytes and astrocytes from the cortical dataset were selected, and those labelled as belonging to the autonomic lineage were selected from the neural crest dataset. Iterative Latent Semantic Indexing (LSI) was performed on the entire dataset and the first 30 dimensions used to integrate the selected cortical, neural crest and EGC data using the Harmony algorithm[85] via the ArchR interface[78]. Clustering was performed using the Louvain algorithm (resolution 1). Two small outlying clusters were removed and iterative LSI, integration using harmony and dimensionality reduction using UMAP were performed on this filtered dataset.

Gene scores were inferred using ArchR[78]; these provide an indication of chromatin acessibility at the gene level. Peaks were called using the "addReproduciblePeakSet()" function in ArchR[86] that calls MACS2[87]. Cells were grouped by cell type for peak calling (autonomic cells, EGCs, oligodendrocytes, astrocytes). To determine the presence/absence of peaks in promoter regions, peaks were called individually for each cell type, using a $q$-value cut-off of 0.05.

To test whether gene modules were enriched in genes with at least one peak in their promoter region we used an upper tail hypergeometric test, with arguments to the R phyper function as follows:

**x** = the number of genes in the gene module with at least 1 peak in their promoter region.

**m** = the number of genes in the whole dataset with at least 1 peak in their promoter region.

**n** = the number of genes in the whole dataset without peaks in their promoter region.

**k** = the number of genes in the gene module.

Pairwise differential gene accessibility (using gene scores) between cell types was calculated using a wilcoxon test in ArchR with "TSSEnrichment" and "log10(nFrags)" as a bias. Here the bias features are quantile normalised, and for each cell in the group to be compared, ArchR selects its closest neighbour in the other group as a background. Pairwise differential peak accessibility was calculated in the same way.

Motif enrichments within differentially accessible peaks (logFC >= 1, $p$adj <= 0.01) were calculated using ArchR for the Homer motif set[86] (http://homer.ucsd.edu/homer/motif/). In addition, chromVAR

deviations[88], bias-corrected measures of how far the per cell accessibility of a motif deviates from the expected norm, were calculated for Homer motifs in ArchR.

## Statistical analysis

Bioinformatics statistical analyses were performed using R and python. All other statistical analyses were performed using GraphPad Prism 9.0 software (GraphPad). Normality distribution was tested using the D'Agostino–Pearson omnibus test. Unpaired two-tailed *t*-tests (followed by Welch's correction test for non-equal standard deviations) were used to compare two groups. When comparing more than two groups, one-way ANOVA followed by Dunnett's multiple-comparison test and Kruskal–Wallis test followed by a Dunn's multiple-comparison test were used for parametric and non-parametric datasets, respectively. $P < 0.05$ was considered to be statistically significant and all of the *P* values are denoted in the figures. The nature of the entity of *n* is defined as individual animals (Fig. 6e and Supplementary Fig. 7l) or field of view (Fig. 6c and Supplementary Fig. 7j) and all experimental replicates are biological. All error bars represent mean ± standard error of the mean (s.e.m.) unless stated otherwise.

## Data visualisation

Data was visualised using GraphPad Prism 9.0 (GraphPad), the python SCANPY[67] and scVelo packages[17], and the R Seurat (v3 and 4)[84], EnhancedVolcano (https://github.com/kevinblighe/EnhancedVolcano), ggplot2 (https://ggplot2.tidyverse.org/) and gplots (https://www.scienceopen.com/document?vid=0e5d8e31-1fe4-492f-a3d8-8cd71b2b8ad9) packages. Figures were created using Adobe Photoshop 2020, Adobe Illustrator 25.3.1, Inkscape 1.1.2 and BioRender.com.

## Reporting summary

Further information on research design is available in the Nature Portfolio Reporting Summary linked to this article.

## Data availability

Raw and processed scRNA-seq data generated in this manuscript are available on the Gene Expression Omnibus (GEO) under the following accessions: GSE237713 (fluidigm scRNA-seq from small intestine SOX10+ cells), GSE237970 (10X scRNA-seq from ganglioid cultures), GSE160196 (bulk RNA-seq from ganglioid cultures), GSE239305 (scATAC-seq data from EGCs), GSE241522 (bulk ATAC-seq data from ganglioid cultures), GSE240190 (scATAC-seq data from trunk/vagal neural crest cells; specific sample GSM7687949), GSE240636 (bulk RNA-seq of EGCs from BAC-treated mice). The following previously generated data used in this manuscript is available on GEO: GSE182715 (fluidigm scRNA-seq from small intestine P60 enteric nervous system cells), GSE129114 (fluidigm scRNA-seq from vagal/trunk neural crest). The previously generated fluidigm scRNA-seq data from small intestine SOX10+ cells is available at www.ebi.ac.uk/arrayexpress/experiments/E-MTAB-5553. scATAC-seq fresh cortical adult mouse brain (P50) data was downloaded from 10X (https://www.10xgenomics.com/resources/datasets?menu%5Bproducts.name%5D=Single%20Cell%20ATAC&page=1&configure%5Bfacets%5D%5B0%5D=chemistryVersionAndThroughput&configure%5Bfacets%5D%5B1%5D=pipeline.version&configure%5BhitsPerPage%5D=500). Adult mouse cortical scRNA-seq data from the Allen Institute for Brain Science was downloaded from https://www.dropbox.com/s/kqsy9tvsklbu7c4/allen_brain.rds. Gene lists used for cell cycle scoring were obtained from Kowalczyk et al.[71]. iCLIP-seq data, which identified direct RNA targets of human CSDE1 in melanoma cells were obtained from Wurth et al.[25]. RNA-seq data on genes differentially expressed in hESCs after CSDE1 knockdown was obtained from Ju Lee et al. (2017)[23]. Human-mouse gene mappings were obtained from the NCBI HomoloGene database (ncbi.nlm.nih.gov/homologene). The GRCm38 genome is available from Ensembl. Source data are provided with this paper. Processed scRNA-seq datasets associated with this manuscript are available to query online (https://bioinformatics.crick.ac.uk/shiny/users/boeings/ENSdevelopment/, https://bioinformatics.crick.ac.uk/shiny/users/boeings/InVitroNeurogenesis/).

## Code availability

TrajectoryGeometry is available as an R Bioconductor package (10.18129/B9.bioc.TrajectoryGeometry) and at https://github.com/AnnaLaddach/TrajectoryGeometry under an MIT License. Other analysis code is available at https://github.com/AnnaLaddach/ABranchingModelOfENSDevelopment under an MIT License.

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

## Acknowledgements

We thank staff at the Crick Science Technology Platforms (STPs) for expert support. In particular, we thank colleagues at the Biological Research Facility, the Flow Cytometry and the Advanced Sequencing Facility. We also thank all members of the Pachnis lab for insightful comments on the manuscript. This work was supported by the Francis Crick Institute, which receives its core funding from Cancer Research UK (FC001128, FC001159), the UK Medical Research Council (FC001128, FC001159) and the Wellcome Trust (FC001128, FC001159). A.L. was partly supported by an Early Career Award from the Wellcome Trust (225712/Z/22/Z). V.P. acknowledges additional funding from BBSRC (BB/L022974) and the Wellcome Trust (212300/Z/18/Z).

## Author contributions

A.L., R.L., S.H.C. and V.P. conceived and designed the study. A.L., R.L., F.P. and V.P. wrote and revised the manuscript. R.L. and F.P. isolated ENS cells from embryonic, postnatal and adult mouse gut. S.H.C. performed the cell culture experiments. E.M.A. and F.P. performed and analysed the BAC treatment experiments. A.C.B.F. assisted with various experiments. M.S.C. and S.U. performed electrophysiology on the in vitro generated neurons. A.E., A.A. and I.A. provided datasets and contributed to the interpretation of bioinformatic analysis. A.L., M.S., S.B. and J.K. performed the bioinformatic and computational analysis. All authors have read and approved the manuscript. R.L., F.P. and M.S. contributed equally to this work and are listed in alphabetical order.

## Funding

## Competing interests

The authors declare no competing interests.
