## [Peer Review File · Nature Communications]

A branching model of lineage differentiation underpinning the neurogenic potential of enteric gliaREVIEWER COMMENTS

Reviewer #1 (Remarks to the Author):

Enteric glia have received a great deal of attention as a potential source of new neurons in adult animals. How the neurogenic potential of enteric glia is regulated remains poorly understood. The study by Laddach and colleagues aims to address this issue by using a combination of single cell transcriptomic profiling, ATACseq to assess open chromatin structure, a novel cell culture system with gene manipulations, and an in vivo gut injury model. The main results suggest that ENS progenitors follow a linear trajectory of gene expression toward a glial default fate or make an early commitment to a neural fate and diverge from the glial path. Interestingly, the data show that enteric glia maintain a memory of their neurogenic past and share chromatin accessibility traits with neural crest precursor cells. Further, in vitro results show that mature glia could reactivate neurogenic transcriptional profiles and form functional neurons in cultures. This ability could be blocked by ablating genes associated with neuronal differentiation and could be triggered in vivo using the BAC injury model. Overall, this is a well-written paper that makes an important contribution to the field. The data are clear, derived from state-of-the-art techniques, and for the most part, convincing. However, the study could be improved by addressing the points noted below.

1) In the sequencing experiments, glia were labeled by a single pulse of tamoxifen and assessed 3 days later for lineage tracing. How long is tamoxifen bioavailable at a concentration sufficient to trigger Cre activity in glia? It would strengthen these experiments to add details on the length of time that the tamoxifen pulse labels cells during development. If the tamoxifen availability window is not tight, it is possible that cells are still being labeled at 2-3 days post injection.

2) Additional details would be beneficial throughout the manuscript. For example, specifying which organ of the small intestine was used and whether sex differences impact the results would be beneficial. In addition, statistical tests, n's, and significance levels are unclear in many figures and legends.

3) A recent paper by Guyer et al (PMID: 36857184) presented single cell sequencing data for enteric glia that suggested the presence of an intraganglionic glial population poised for neurogenesis based on maintaining accessible chromatin. This paper should be discussed in relation to the results presented here.

4) For neurogenic cultures, can the authors be confident that neurons are derived from "enteric glia" rather than precursor cells within ganglia that are captured by the labeling and isolation technique? Also, are Schwann cell precursors isolated in this technique and contribute to in vitro neurogenesis?

5) I'm not sure the BAC experiment is that convincing. It seems that only two genes are upregulated in this model that correspond to module 16 and one is actually downregulated. It may be beneficial to highlight numbers of genes associated with the GO terms in panel t to give a better idea of the extent of gene changes. It would also be beneficial to describe how many glia could be isolated in this model and whether these are actually enteric glia or Schwann cell precursors that migrate to the gut.

6) The final section “Cell state transitions...” doesn’t really seem to add as much to this paper as other sections. It may be better to move this figure to a supplement to allow space to split and expand some of the earlier, complex figures.

7) The rigor of the experiments shown in Figure 5 seems relatively low if my understanding of the n’s is correct. It seems like the n’s here just represent fields of view in only two cell cultures (?). It also isn’t clear if these cultures are derived from the same animal.

8) It is necessary to abbreviate enteric glia as EGC? There are several similar abbreviations in the ENS field (ENS, ENC, EEC, ECC, etc) and it only takes one more letter to spell glia. It could help to reduce confusion with similar abbreviations if this abbreviation was not used.

9) Figure 1c: How is cluster 6 defined as neurons if they also express significant levels of Sox10, Plp1, S100B, and GFAP?

10) Figure 1d: Can the authors provide more explanation for why there is a greater proportion of progenitors at P1 than at earlier embryonic stages?

11) Figure 1f: There seems to be a very early branch point (orange cells) that is undefined. It is clear that neurons diverge later and that glia remain along the same trajectory but additional clarification of what this other branch is would be beneficial.

12) Figure 1g-j: This analysis was a bit difficult to understand and it would help to add a bit more in the main text to help to explain the significance of the method and results.

13) Figure 2d-e: These immunolabeling panels are very small and over saturated. It would be good to expand the size of these, also show overlays, and limit red/green combinations when possible.

14) The data in Figure 4a show very nice differences in chromatin accessibility between enteric glia, astrocytes, and oligodendrocytes. How do overall gene expression patterns differ between astrocytes and enteric glia in the scRNAseq data?

15) Figure 5d-f & m: These are really tiny images that are hard to see what is shown by the labeling. It would be great if these could be enlarged (maybe another figure in place of current figure 6?) so the labeling could be viewed clearly.

16) Figure 5j: Can the authors please explain why the branching model comes out different here than during development? I would have expected that glia revert to progenitor-like cells and then follow the same trajectory and branching as before but this doesn’t seem to be the case.

17) Page 3, line 2: Please revise “neuron” to “neurons”

18) Figure S1c: Can the arrows showing directionality be enlarged here? Difficult to see.

19) Figure S4: These panels are very small and over saturated. Can these be enlarged and adjusted to show the data clearly?

Reviewer #2 (Remarks to the Author):

This study aims to understand the molecular mechanisms underlying the neurogenic potential of enteric glial cells (EGCs) using single-cell transcriptomic and epigenetic profiling, gene targeting in a cell culture system, and a gut injury model. The authors conclude that neural crest cells follow a linear default trajectory of gliogenic differentiation as they progressively lose their proliferative and neurogenic biases and become quiescent, mature EGCs. However, along that progenitor-to-glia axis, the cells retain neurogenic potential by maintaining open chromatin for neurogenic gene promoters, allowing EGCs to undergo neuronal differentiation utilizing the same differentiation programs employed by the embryonic ENS. The study provides a lot of valuable data and presents an interesting perspective on the transcriptional landscape underlying ENS development and the neurogenic potential of glia. However, there are several major concerns as detailed below.

1. The paper relies heavily on computational modeling of single-cell transcriptional data. While the authors do a thorough job describing the methodology, there is little validation of the conclusions derived from these models, especially with respect to embryonic development. The claims made in the manuscript are thus overly strong and should either be supported by experimental validation or stated in a manner that makes clear their speculative nature. For example, a major conclusion of the paper is to propose a lineage trajectory of how neural crest cells differentiate, but it's based on a mathematical model that utilizes transcriptional snapshots at specific timepoints. Fig. 1F visually depicts the proposed differentiation, but there is no experimental evidence to support this. Testing this experimentally may be challenging, but at the very least the conclusions need to be tempered.
2. Some of the main findings in this study are not novel and have previously been reported in the literature, including several recent papers (Morarach et al, Nat Neurosci, 2021; Guyer, Cell Reports, 2023). The authors should more clearly state the novelty and significance of their findings in relation to the existing literature, and cite and discuss these recent papers in light of their current work.
3. The prose is very dense and difficult to follow, even for individuals with experience in the field. The authors should consider revising the language to make the manuscript more accessible to a broader scientific audience.
4. It is unclear exactly what biological claims the authors are seeking to make. Since their model of embryonic development is not experimentally validated, it is not clear whether or not it provides valid biological insights. Their use of a cell culture system lends some evidence to their model's validity, but this artificial system cannot provide evidence for their claims regarding embryonic development. The paper would be improved if a specific biological hypothesis were clearly stated in the Abstract and Introduction, and their results were discussed in relation to that hypothesis.
5. In the Results section titled "Neurogenic cultures of EGCs," it states that within 4 DIV, "EGCs acquired the morphology of early neural crest cells...." but no information is provided on what this morphology is nor is an image included to support the statement. The figure panel only shows staining for Sox10, EdU, and pH3, all of which are nuclear and therefore no cell morphology can be visualized.
6. In Fig. 4, epigenetic profiling is compared between ANCCs and EGCs. The ANCC single cell data appears to be from E9.5 trunk neural crest, which does not include the vagal or sacral crest that gives rise to enteric neurons and EGCs. This should be acknowledged and how this may impact the results should be discussed.
7. Fig. 5J refers to "glia," "glial-like," and "progenitor-like." An explanation needs to be

provided for how each of these terms is defined.

8. The BAC model is used in Fig. 5 to injure the ENS, but no histology or immunohistochemistry is provided to confirm that the model worked as expected. Furthermore, Fig. 5U shows only 4 genes with a >2-fold change in expression. This suggests the model did not work since, if it did, one would expect many more than just a few genes to be differentially expressed.

9. Fig. 6 does not seem to fit into the manuscript and does not add much. Based on this figure, the authors state that “common transcriptional mechanisms” are shared by the CNS and PNS. That is a big claim based on the data shown, which is limited to several gene expression modules. This figure can either be removed or moved to supplementary data.

10. Spelling of “benzalkonium” (bottom of page 9).

Reviewer #3 (Remarks to the Author):

By performing comprehensive single cell gene expression analyses of neural crest cells and their progenies through development and adult stages, Laddach et al. have defined cell state dynamics in both neuronal and glial lineages. It is known that enteric glial cells exhibit a surprising cell plasticity, enabling neurogenesis upon injury. The authors' single cell chromatin analysis revealed that chromatin remain open for neurogenic loci of early enteric nervous system progenitors in mature glial cells, providing new, important mechanistic insight underlying glial cell plasticity. Moreover, their genetic and glial cell culture studies further support the finding that neuronal differentiation of glia is driven by transcriptional programs of early enteric nervous system progenitors. As enteric glial cells are also implicated in gut stem cell homeostasis and regeneration, this well executed study would likely make a broad impact in both the gut biology and neuroscience fields. The following suggestions considered to strengthen an already strong manuscript.

The last sentence of paragraph 1 on page 5 states “Together, our experimental and computational analyses suggest that during mammalian development ENS progenitors either commit to neurogenic paths or proceed along a linear default differentiation pathway giving rise eventually to mature enteric glia.” It is unclear whether the authors are claiming that the default lineage state of ENS progenitors is neuronal or glial. If so, are the transcriptional programs of ENS progenitors similar to neuronal or glial lineage specification programs (instead of comparing them to mature adult neuron and glial cell gene expression)? If the authors didn't mean to address the default state, they could simply remove “default”.

The authors' genetic and glial cell culture studies demonstrate that *Ascl1* and *Foxd3* are required for the neuronal differentiation of glial cells. Could the authors investigate their available (published) genomic binding sites with the ATAC-seq data? Finding their binding sites in chromatin regions that remained open in EGCs would highlight the mechanisms of their efficient reprogramming for neuronal differentiation; found in closed chromatin regions, they might suggest a requirement for the pioneering activity of *Ascl1* and/or *Foxd3*. This result could also explain why neuronal differentiation of glial cells is extremely inefficient without injury or other stimulation.

The last section addresses the cell state transition of enteric nervous system progenitors in comparison to neural stem cells. While this part is interesting, it would require further analysis. Go term and gene set enrichment analyses should be performed to further define

the transcriptional programs conserved in both systems. What specific genes and/or signaling pathways might be involved?

Response to Reviewers

Reviewer 1

1) In the sequencing experiments, glia were labeled by a single pulse of tamoxifen and assessed 3 days later for lineage tracing. How long is tamoxifen bioavailable at a concentration sufficient to trigger Cre activity in glia? It would strengthen these experiments to add details on the length of time that the tamoxifen pulse labels cells during development. If the tamoxifen availability window is not tight, it is possible that cells are still being labeled at 2-3 days post injection.

We thank the referee for the opportunity to clarify this important point. Published reports have demonstrated that a single dose of tamoxifen induces rapid (within 24 hours) recombination-mediated activation of a reporter in mouse embryos ¹. A separate study demonstrated that effective tamoxifen levels for Cre activation were restricted to 12 hours ². In a recent seminar evidence was presented that the half-life of tamoxifen is 2-4 hours (although we are currently unable to confirm whether this information is published). These reports are in agreement with our earlier studies in which we used the Sox10CreER^{T2} transgene in conjunction with the Rosa26-Confetti reporter to sparsely label ENS progenitors by exposing E12.5 mouse embryos to a single dose of tamoxifen ³. Strikingly, the number of Confetti⁺ ENS clones identified 20-24 hours later (E13.5) remained unchanged throughout the embryonic and postnatal stages we analyzed (including adult), indicating that all founder cells were labelled within the first 24 hours after tamoxifen administration. Finally, and most importantly, as shown in Fig. 1e of our manuscript, all cells labelled at E12.5 and traced for 3 days were part of the E17.5 cluster. If this “relocation” of the labelled cells to the late clusters resulted from continued recombination by residual tamoxifen, we would have expected that at least some (and most likely most) of the labelled cells would remain within the early cluster.

2) Additional details would be beneficial throughout the manuscript. For example, specifying which organ of the small intestine was used and whether sex differences impact the results would be beneficial. In addition, statistical tests, n's, and significance levels are unclear in many figures and legends.

We have added more experimental details in our revised manuscript. Specifically, in response to the reviewer's questions we state clearly in the Methods section that for all pre- and perinatal stages (E12.5, E16.5, P0) tdT⁺ cells were isolated from the entire small intestine (from duodenum to terminal ileum). For the P25 and adult stages, tdT⁺ cells were isolated from the muscularis externa of the small intestine (duodenum to terminal ileum).

All transcriptomic experiments were performed with equal numbers of male and female mice. Also, our bioinformatic analysis indicated that cells from the same time point did not cluster on the basis on Xist expression (indicative of female cells), suggesting lack of major sex-dependent transcriptional differences.

We have also added cell numbers for transcriptomic experiments to the figure legends. In Fig. 1, panels h and i, asterisks have been replaced by p values to denote significance. P values for Supplementary Fig.

2b are now provided in Supplementary Tables 1-3. The details of all statistical tests have been added in the figure legends.

3) A recent paper by Guyer et al (PMID: 36857184) presented single cell sequencing data for enteric glia that suggested the presence of an intraganglionic glial population poised for neurogenesis based on maintaining accessible chromatin. This paper should be discussed in relation to the results presented here.

We thank the referee for pointing this out. The Guyer et al. paper was published while our manuscript was under review and unfortunately, we did not have the opportunity to discuss its findings. Although the starting point and focus of the two studies are different, we are very pleased that data presented in the Guyer et al. paper are consistent and supportive of our own findings, such as the observation that EGCs maintain an open chromatin configuration at neurogenic loci. We note however, that in the Guyer et al. paper many conclusions are based on RNA-seq/ATAC-seq datasets from cells isolated from neurospheres or P14 mice, both of which are non-steady state systems. Nevertheless, we are delighted to have the opportunity to refer to and discuss the Guyer et al. paper in our revised manuscript.

4) For neurogenic cultures, can the authors be confident that neurons are derived from “enteric glia” rather than precursor cells within ganglia that are captured by the labeling and isolation technique? Also, are Schwann cell precursors isolated in this technique and contribute to *in vitro* neurogenesis?

We thank the reviewer for raising these important points. Our neurogenic/ganglioid cultures are established from Sox10⁺ cells of the longitudinal muscle/myenteric plexus (LMMP) layer. A series of published experiments (and additional unpublished work from our lab), including immunostaining for proliferative markers, EdU incorporation experiments and extensive transcriptomic analysis, have demonstrated conclusively that enteric glial cells are generally quiescent at steady state (further support is now provided by the Guyer et al. paper). In addition, both the Fluidigm and 10X scRNA-seq datasets generated and analyzed in this manuscript did not identify distinct clusters with progenitor characteristics. Therefore, it is unlikely that the neurons in ganglioid cultures originate from a “proliferative neurogenic precursor subtype” of enteric glial cells. Instead, our combined lineage tracing and transcriptomic analysis argue strongly that the tdT⁺ glial cells when placed in culture give rise to proliferative precursors which downregulate gene expression programs associated with mature EGCs, acquire properties of ENS progenitors and undergo neuronal differentiation. However, it is unclear whether all glial cells within our starting population have neurogenic potential or whether this is a property characteristic of a subset of non-neuronal cells of the myenteric plexus. The Guyer et al. paper suggests that the neurogenic potential is a specific property of ganglionic glia, but our own studies do not address this question directly.

Regarding the potential contribution of Schwann cell precursors (SCPs) to *in vitro* neurogenesis, we are currently unable to answer this question directly. However, given the relatively small contribution of SCP-derived cells to the mouse ENS and the lineage tracing experiments (mainly from the Enomoto lab using Dhh-Cre transgenics) demonstrating that in the small intestine of wild-type animals SCPs contribute mainly to the submucosal plexus and deep layers^{4,5}, we argue that the starting population of

Sox10⁺ cells in the ganglioid cultures, which were established from myenteric plexus preparations, contains very few derivatives of Schwann cells. The sparsity of SCP-derived cells in the myenteric plexus layer was one of the main reasons we chose longitudinal muscle-myenteric plexus (LMMP) preparations for our ganglioid cultures. In addition, the Enomoto lab has demonstrated that SCP-derived neurogenesis is independent of RET signaling^{4,5}, while our CRISPR experiments have clearly shown that ablation of this pathway blocks neuronal development in ganglioid cultures. Finally, and despite the widespread use of Schwann cells cultures in many studies of peripheral nerve injury, to our knowledge no published reports have demonstrated that adult Schwann cells are capable of generating neurons in culture.

5) I'm not sure the BAC experiment is that convincing. It seems that only two genes are upregulated in this model that correspond to module 16 and one is actually downregulated. It may be beneficial to highlight numbers of genes associated with the GO terms in panel t to give a better idea of the extent of gene changes. It would also be beneficial to describe how many glia could be isolated in this model and whether these are actually enteric glia or Schwann cell precursors that migrate to the gut.

We thank the reviewer for giving us the opportunity to clarify this point. We have now revised our manuscript and present in Fig. 6h a new volcano plot which clearly shows that many genes are upregulated in response to BAC treatment. The upregulated genes are color coded to indicate the GO terms they correspond to (shown in revised Fig. 6g). This panel also includes the number of genes associated with the GO terms, as requested by the reviewer. The reviewer correctly points out that it is only two genes (*Hmga2* and *Igfbp2*) from GM16 that are clearly upregulated in this experimental model. Please note that in this model gene expression is analyzed only 48 hours after injury and therefore it is biased towards early responding genes. In this regard, it is interesting that *Hmga2* and *Igfbp2* are the most highly upregulated genes in enteric glia-derived ganglioid cultures 4 days after plating (Fig. 5j). Also, in response to a comment by reviewer 2 we have now added in our revised manuscript a Supplementary Figure which describes in further detail the proliferative response of enteric glia to BAC treatment. We also provide the number of sequenced glial cells from control and BAC-treated animals in the relevant Methods section of our manuscript.

Regarding the potential of Schwann cells migrating to the gut in response to BAC treatment, the tools used in our current experiments do not allow us to answer this question directly.

6) The final section "Cell state transitions..." doesn't really seem to add as much to this paper as other sections. It may be better to move this figure to a supplement to allow space to split and expand some of the earlier, complex figures.

We agree with the referee, although this section is very interesting, it does not add substantially to the manuscript. Therefore, in response to the reviewer's comment (and a similar point raised by the other reviewers), we have removed this data from the revised manuscript.

7) The rigor of the experiments shown in Figure 5 seems relatively low if my understanding of the n's is correct. It seems like the n's here just represent fields of view in only two cell cultures (?). It also isn't clear if these cultures are derived from the same animal.

We thank the reviewer for giving us the opportunity to clarify our experimental design. The data shown in the original Fig. 5 m,n (and in the revised manuscript in Fig. 6b,c) are based on two experiments. For each experiment, EGCs were derived from 3 mice. As per our protocol, each EGC culture was split at DIV4 into three wells per condition (control and CRISPRs). n's represent equal number of fields of view from both cultures. It is important also to mention that the results of both the Ret and Ascl1 CRISPR experiments were supported by orthogonal approaches. Thus, similar to the Ret CRISPR results, we observed that pharmacological inhibition of Ret also leads to reduced neuronal differentiation in EGC-derived ganglioids (Supplementary Fig. 7h-j). In addition, in an experiment analogous to that shown in the original Fig. 5 o,p (now in revised Fig. 6d,e), in which ganglioids were established from Foxd3-deficient glial cells, we have also generated ganglioid cultures from mice in which *Ascl1* was conditionally deleted in vivo from EGCs. The results of one experiment are consistent with the *Ascl1* CRISPR experiment shown in revised Fig. 6a-c.

8) It is necessary to abbreviate enteric glia as EGC? There are several similar abbreviations in the ENS field (ENS, ENC, EEC, ECC, etc) and it only takes one more letter to spell glia. It could help to reduce confusion with similar abbreviations if this abbreviation was not used.

We understand the frustration of the reviewer with the confusing abbreviations. However, amongst all these similar looking/sounding abbreviations, ENS and EGCs are the most established and universally recognized and, if the reviewer agrees we would prefer to stick with both in our manuscript.

9) Figure 1c: How is cluster 6 defined as neurons if they also express significant levels of Sox10, Plp1, S100B, and GFAP?

Fig. 1c does not show that cluster 6 expresses statistically significant levels of these genes; instead, this panel simply indicates the mean level of gene expression within a cluster and the fraction of cells that express a gene. Expression levels are in fact significantly lower than in enteric glia (clusters 7 and 8). The results of a Wilcoxon test are reported below:

Gene	C6 v C7 LogFC	C6 v C8 LogFC	C6 v C7 Padj	C6 v C8 Padj
Sox10	-5.705466	-5.4826975	2.46E-13	6.41E-05
Plp1	-6.179948	-3.900884	1.70E-15	1.84E-03
S100b	-4.165313	-5.833002	3.78E-10	2.68E-06
Gfap	-2.466428	-9.367997	5.68E-03	8.05E-06

It should be noted that our scRNAseq dataset indicates expression of the transcript rather than the presence of its protein product (detected in the numerous immunostaining experiments reported in the literature), and enteric neurons also show expression of these genes in other published datasets⁶. Moreover, co-expression (identified by immunostaining) of the neuronal marker Hu and S100b or

Sox10 has been reported at both 2 and 6 weeks postnatally ⁷. Finally, markers for this cluster include neuropeptides. A few examples are given in the table below:

Gene	LogFC	Padj
Nos1	6.710251	3.36E-17
Chat	7.613453	1.70E-12
Calb2	10.01886	5.77E-21

10) Figure 1d: Can the authors provide more explanation for why there is a greater proportion of progenitors at P1 than at earlier embryonic stages?

Our previous lineage tracing experiments have demonstrated that the fraction of adult enteric neurons generated from Sox10⁺ ENS progenitors is highest at mid-gestation, reduced at perinatal stages and extinguished in adulthood ⁸. Our current transcriptomic analysis, indicating reduced representation of neuronally committed cells within the tdT⁺ ENS cell population from P1 gut relative to earlier stages, is in agreement with these findings. We argue therefore that the neurogenic output of progenitors (as defined by expression of the ENS progenitor marker genes *Sox10*, *ErbB3*, *Fabp7* and *Plp1*) decreases during development.

11) Figure 1f: There seems to be a very early branch point (orange cells) that is undefined. It is clear that neurons diverge later and that glia remain along the same trajectory but additional clarification of what this other branch is would be beneficial.

This is also a neurogenic branch and shows neurons being born at E13.5. It corresponds to the early neurogenic branch emerging from early ENS progenitors, also depicted in the inset of Fig. 3c. We have added in our revised manuscript labels to Fig. 1f to make it clearer.

12) Figure 1g-j: This analysis was a bit difficult to understand and it would help to add a bit more in the main text to help to explain the significance of the method and results.

In response to the referee's comment, we have edited the relevant part of the main text. We hope that our revisions together with the extended description of this package in the Methods and in <https://bioconductor.org/packages/devel/bioc/html/TrajectoryGeometry.html> will help the audience to understand the principle and applications of the package.

13) Figure 2d-e: These immunolabeling panels are very small and over saturated. It would be good to expand the size of these, also show overlays, and limit red/green combinations when possible.

In response to the reviewer's comment, we have now expanded the size of the images shown in the revised Fig. 2d,e. The resolution of the images is such that zooming into the panels would allow the

reader to see clearly the details of the immunostainings. In addition, we now show overlays and combine green with magenta, rather than red.

14) The data in Figure 4a show very nice differences in chromatin accessibility between enteric glia, astrocytes, and oligodendrocytes. How do overall gene expression patterns differ between astrocytes and enteric glia in the scRNAseq data?

We are currently analyzing the transcriptomic differences between enteric glial cells (EGC), astrocytes (Astro) and oligodendrocytes (Oligo) in detail. We are of the opinion that this analysis is outside the scope of our manuscript and therefore we present for the benefit of the reviewer a UMAP of the transcriptomes of the three glial cell types and a dot blot of the top 10 differentially expressed genes, which demonstrate the unique character of enteric glia.

15) Figure 5d-f & m: These are really tiny images that are hard to see what is shown by the labeling. It would be great if these could be enlarged (maybe another figure in place of current figure 6?) so the labeling could be viewed clearly.

We have now revised our manuscript according to the reviewer's suggestion. We have also split the original Fig. 5 into two figures (Fig. 5 and Fig. 6) and enlarged the image sizes.

16) Figure 5j: Can the authors please explain why the branching model comes out different here than during development? I would have expected that glia revert to progenitor-like cells and then follow the same trajectory and branching as before but this doesn't seem to be the case.

We thank the reviewer for the opportunity to clarify this point. UMAP representations (used for the ganglioid scRNA-seq data shown in the original and revised Fig. 5l) are non-linear and therefore do not describe geometry. The branching geometry of the *in vivo* data was only revealed when PCA analysis was used (Fig. 1f). Using a PCA plot for the *in vitro* ganglioid transcriptomic analysis also revealed a similar branching geometry, with cells of the gliogenic lineage forming a "backbone" from which the neurogenic trajectory branches off. For clarity, we have now added this PCA plot in the revised manuscript (Fig. 5n).

17) Page 3, line 2: Please revise "neuron" to "neurons"

We have corrected this.

18) Figure S1c: Can the arrows showing directionality be enlarged here? Difficult to see.

We have enlarged the whole Supplementary Fig. 1c to make it easier to see.

19) Figure S4: These panels are very small and over saturated. Can these be enlarged and adjusted to show the data clearly?

In response to the reviewer's comment, we have now enlarged these panels and also use magenta instead of red.

Reviewer 2

1. The paper relies heavily on computational modeling of single-cell transcriptional data. While the authors do a thorough job describing the methodology, there is little validation of the conclusions derived from these models, especially with respect to embryonic development. The claims made in the manuscript are thus overly strong and should either be supported by experimental validation or stated in a manner that makes clear their speculative nature. For example, a major conclusion of the paper is to propose a lineage trajectory of how neural crest cells differentiate, but it's based on a mathematical model that utilizes transcriptional snapshots at specific timepoints. Fig. 1F visually depicts the proposed differentiation, but there is no experimental evidence to support this. Testing this experimentally may be challenging, but at the very least the conclusions need to be tempered.

4. It is unclear exactly what biological claims the authors are seeking to make. Since their model of embryonic development is not experimentally validated, it is not clear whether or not it provides valid biological insights. Their use of a cell culture system lends some evidence to their model's validity, but this artificial system cannot provide evidence for their claims regarding embryonic development. The paper would be improved if a specific biological hypothesis were clearly stated in the Abstract and Introduction, and their results were discussed in relation to that hypothesis.

We thank the reviewer for their comments and hope that our response provides some clarity and addresses their concerns as presented in the related comments 1 and 4.

We disagree with the reviewer that our “paper relies heavily on computational modeling of single-cell transcriptional data”. We do not consider our work as “computational modelling”. The novel R package (TrajectoryGeometry) and associated analysis (presented in 4 panels of Fig. 1 and in Supplementary Fig. 2) are used to quantify experimentally generated transcriptomic data (as countless algorithms and packages do nowadays) and the values we obtained led us to formulate a novel (and in our view exciting) model of ENS lineage configuration that highlights the differential degree of gene expression change associated with neurogenesis and gliogenesis in the ENS. Thus, rather than coming up with a computer-generated model in need of experimental validation, we are instead proposing a model that is based on experimental data and provides a framework for understanding a series of observations in the field of enteric neuroscience (including the striking paucity of molecular markers that distinguish adult enteric glia from ENS progenitors and the neurogenic potential of enteric glia). No other part of our manuscript relies on the proposed model. Instead, most of our work describes experiments, which in addition to supporting the lineage configuration model, also provide novel mechanistic insight into ENS lineage development and function.

As is true for all models that aim to provide mechanistic understanding of biological processes, our ENS lineage differentiation model could make further predictions and enable us to design experiments which are likely to generate additional valuable information. We agree with the reviewer that our model “utilizes transcriptional snapshots at specific timepoints” and that more needs to be done to “test our model” and uncover mechanisms that underpin the dynamics of lineage differentiation in the ENS. In our view, a direct test would involve the recording of transcriptomic changes of ENS progenitors in real developmental time. However, given the relative inaccessibility of the mammalian ENS, this work would be more feasible in a different experimental system, such as zebrafish. We aim to address these questions in the future but we believe that they are beyond the scope of the current manuscript.

Our current work was initiated in order to address fundamental questions emerging from previously published studies (from several groups including ours). Although an earlier publication from our lab identified unequivocally Sox10⁺ cells in mouse embryo gut as bipotential progenitors to mature enteric neurons and glia³, it did not examine the intermediate steps, thus leaving a gap in our understanding of how ENS lineage differentiation unfolds during development. Our current experiments aim at filling this gap and provide novel mechanistic insight into the cellular and molecular processes underpinning the formation of effector ENS lineages (neurons and glia). In doing so, it also furnishes a developmental framework for understanding a fundamental property of vertebrate enteric glia, namely the ability to activate their intrinsic neurogenic potential under *in vivo* (zebrafish) or experimental (mouse) conditions

Specifically, the aim of our study was to identify transcriptomic profiles that reflect cell state transitions along both the gliogenic and neurogenic ENS lineages, including committed progenitor states. Although this work led to the identification of transcriptional profiles associated with committed neuronal progenitors, we obtained no evidence that committed glial precursor states were distinct from bipotential progenitors. Rather, mature glia formed a continuum with progenitor cells, indicating that no point of commitment exists. This agrees with our *in vitro* data demonstrating that glial cells can give rise to neurons, but also *in vivo* data demonstrating that glia can undergo limited neurogenesis in response to injury in mammals but engage in constitutive neurogenesis in teleosts. Consistent with these observations, as far as we are aware, no genetic manipulation is known that selectively targets gliogenesis without affecting neuronal differentiation and, as mentioned in our manuscript, all cases of Hirschsprung's disease are characterized by local elimination of both neurons and glial cells. These observations further support the absence of a separate gliogenic process that is distinct from the time-axis aligned maturation of ENS progenitors.

2. Some of the main findings in this study are not novel and have previously been reported in the literature, including several recent papers (Morarach et al, Nat Neurosci, 2021; Guyer, Cell Reports, 2023). The authors should more clearly state the novelty and significance of their findings in relation to the existing literature, and cite and discuss these recent papers in light of their current work.

We thank the reviewer for pointing out these two papers. We follow closely the work of the Marklund lab (Dr Marklund was a postdoc in the laboratory of the corresponding author), but the focus in the Morarach et al. paper is quite distinct from our manuscript. However, in response to the referee's comment we now refer to their paper in the Results section of our manuscript "Shared transcriptional programs during early and late ENS neurogenesis".

Regarding the referee's comment about the Guyer et al. paper, please see our response to a similar comment (#3) by Reviewer 1.

3. The prose is very dense and difficult to follow, even for individuals with experience in the field. The authors should consider revising the language to make the manuscript more accessible to a broader scientific audience.

We are sorry the reviewer found our text "difficult to follow". We admit that our text is somewhat dense (most likely due to the use of a broad range of experimental and bioinformatic tools and the length restrictions) but several colleagues (working on unrelated fields) and Reviewers 1 and 3, found our manuscript clear. We hope that our revisions in the Results, Discussion and Methods sections will provide further clarity.

5. In the Results section titled "Neurogenic cultures of EGCs," it states that within 4 DIV, "EGCs acquired the morphology of early neural crest cells..." but no information is provided on what this morphology is nor is an image included to support the statement. The figure panel only shows staining for Sox10, EdU, and pH3 pH3, all of which are nuclear and therefore no cell morphology can be visualized.

We thank the reviewer for raising this point. We agree with them that we do not provide direct evidence that DIV4 EGC-derived cells in culture “acquired morphology of neural crest cells”. However, these cells have striking morphological similarities to ENS progenitors that are generated in culture from embryonic and postnatal gut tissue and characterized previously by our group and other labs^{10,11}. Therefore, and in order to be more accurate, we have now changed the text stating that “EGCs acquired the morphology of early ENS progenitors”.

6. In Fig. 4, epigenetic profiling is compared between ANCCs and EGCs. The ANCC single cell data appears to be from E9.5 trunk neural crest, which does not include the vagal or sacral crest that gives rise to enteric neurons and EGCs. This should be acknowledged and how this may impact the results should be discussed.

As described in the Methods section, the autonomic neural crest scATAC-seq dataset was generated from E9.5-10.5 *Wnt1Cre-Rosa26-tdTomato* embryos. The head of the embryos was removed with a razor blade anterior to the otic vesicle and Tomato⁺ cells were isolated from the remaining part of the embryo. Therefore, the starting cell population includes derivatives of both the vagal and trunk neural crest, including autonomic neural crest cells (ANCCs). To the best of our knowledge, there is no evidence to suggest that trunk and vagal autonomic neural crest are distinct. In fact, previous published evidence, including earlier work from our group, has demonstrated that the ENS develops from a mixture of vagal and trunk neural crest cells¹².

7. Fig. 5J refers to “glia,” “glial-like,” and “progenitor-like.” An explanation needs to be provided for how each of these terms is defined.

We thank the reviewer for the opportunity to clarify these labels. We use the term glia to refer to glial cells freshly isolated from their *in vivo* environment. We use the term progenitor-like to refer to glial cells *in vitro* that have acquired a transcriptomic profile similar to ENS progenitor cells. We use the term glial-like to refer to cells in ganglioid cultures that express glial markers (such as S100b) but lack expression of gene modules (such as GM 75) that characterize mature glial cells *in vivo*. This is now explained clearly in the text and the Figure legend of the revised Fig. 5.

8. The BAC model is used in Fig. 5 to injure the ENS, but no histology or immunohistochemistry is provided to confirm that the model worked as expected. Furthermore, Fig. 5U shows only 4 genes with a >2-fold change in expression. This suggests the model did not work since, if it did, one would expect many more than just a few genes to be differentially expressed.

We thank the referee for raising this point. In response to their comment, we have now revised our manuscript and added a new Figure (Supplementary Fig. 8), which shows immunostainings and quantification of proliferating glial cells demonstrating that the BAC treatment worked as expected.

Our aim in Fig. 5u was to depict specifically GM16 genes that showed significant changes in expression. However, many more genes were up- and down-regulated as a result of the BAC treatment. In response to the referee’s comment (and also a similar comment by Reviewer 1), we now include a volcano plot that shows all differentially expressed genes between control and BAC-treated tissue (revised Fig. 6h).

Genes for module GM16 are color-coded to correspond to GO terms shown in Fig. 5g. Please, see also response to comment #5 by Reviewer 1.

9. Fig. 6 does not seem to fit into the manuscript and does not add much. Based on this figure, the authors state that “common transcriptional mechanisms” are shared by the CNS and PNS. That is a big claim based on the data shown, which is limited to several gene expression modules. This figure can either be removed or moved to supplementary data.

In response this comment, and similar comments by the other reviewers, this data has now been removed from the manuscript. Please see also our response to comment #6 of Reviewer 1.

10. Spelling of “benzalkonium” (bottom of page 9).

This spelling mistake has been corrected.

Reviewer 3

1. The last sentence of paragraph 1 on page 5 states “Together, our experimental and computational analyses suggest that during mammalian development ENS progenitors either commit to neurogenic paths or proceed along a linear default differentiation pathway giving rise eventually to mature enteric glia.” It is unclear whether the authors are claiming that the default lineage state of ENS progenitors is neuronal or glial. If so, are the transcriptional programs of ENS progenitors similar to neuronal or glial lineage specification programs (instead of comparing them to mature adult neuron and glial cell gene expression)? If the authors didn’t mean to address the default state, they could simply remove “default”.

We thank the reviewer for giving us the opportunity to clarify this point. We argue here that the default state of ENS progenitors is glial and that, contrary to the neurogenic trajectory, there are no obvious points of commitment of glial cells along the gliogenic trajectory. Therefore, our data suggests there is a single decision ENS progenitors are faced with, namely to become a neuron or not to become a neuron. Consequentially, gliogenesis can be considered a default path. We have added this argument to the main text.

2. The authors’ genetic and glial cell culture studies demonstrate that *Ascl1* and *Foxd3* are required for the neuronal differentiation of glial cells. Could the authors investigate their available (published) genomic binding sites with the ATAC-seq data? Finding their binding sites in chromatin regions that remained open in EGCs would highlight the mechanisms of their efficient reprogramming for neuronal differentiation; found in closed chromatin regions, they might suggest a requirement for the pioneering activity of *Ascl1* and/or *Foxd3*. This result could also explain why neuronal differentiation of glial cells is extremely inefficient without injury or other stimulation.

We thank the reviewer for their very interesting comment and suggestion. We have looked into *Ascl1* and *Foxd3* motif enrichment and present here our preliminary analysis. As expected, *Ascl1* binding sites are enriched in autonomic neural crest cells (ANCCs) but also in oligodendrocytes (Oligo), consistent with the role of this transcription factor in oligodendroglia differentiation. Interestingly, *Ascl1* motifs are also found in a subset of EGCs, indicating perhaps their enhanced neurogenic potential. However, it is currently unclear whether this subset of enteric glia corresponds to a region-specific subpopulation. Given that in this analysis we only characterize sites of potential binding for *Ascl1* and *Foxd3* rather than actual binding, we believe that it is appropriate to provide this information only for the benefit for the reviewer.

3. The last section addresses the cell state transition of enteric nervous system progenitors in comparison to neural stem cells. While this part is interesting, it would require further analysis. Go term and gene set enrichment analyses should be performed to further define the transcriptional programs conserved in both systems. What specific genes and/or signaling pathways might be involved?

We agree with this referee that this part requires further analysis which would be beyond the scope of the current work. In response to this comment and similar comments by the other referees, we have decided to remove this part from this manuscript.

References

- 1 Danielian, P. S., Muccino, D., Rowitch, D. H., Michael, S. K. & McMahon, A. P. Modification of gene activity in mouse embryos in utero by a tamoxifen-inducible form of Cre recombinase. *Current biology : CB* **8**, 1323-1326, doi:10.1016/s0960-9822(07)00562-3 (1998).

- 2 Nakamura, E., Nguyen, M. T. & Mackem, S. Kinetics of tamoxifen-regulated Cre activity in mice using a cartilage-specific CreER(T) to assay temporal activity windows along the proximodistal limb skeleton. *Dev Dyn* **235**, 2603-2612, doi:10.1002/dvdy.20892 (2006).
- 3 Lasrado, R. *et al.* Lineage-dependent spatial and functional organization of the mammalian enteric nervous system. *Science* **356**, 722-726, doi:10.1126/science.aam7511 (2017).
- 4 Uesaka, T., Nagashimada, M. & Enomoto, H. Neuronal Differentiation in Schwann Cell Lineage Underlies Postnatal Neurogenesis in the Enteric Nervous System. *J Neurosci* **35**, 9879-9888, doi:10.1523/JNEUROSCI.1239-15.2015 (2015).
- 5 Uesaka, T. *et al.* Enhanced enteric neurogenesis by Schwann cell precursors in mouse models of Hirschsprung disease. *Glia* **69**, 2575-2590, doi:10.1002/glia.24059 (2021).
- 6 Zeisel, A. *et al.* Molecular Architecture of the Mouse Nervous System. *Cell* **174**, 999-1014.e1022, doi:10.1016/j.cell.2018.06.021 (2018).
- 7 Parathan, P., Wang, Y., Leembruggen, A. J., Bornstein, J. C. & Foong, J. P. The enteric nervous system undergoes significant chemical and synaptic maturation during adolescence in mice. *Dev Biol*, doi:10.1016/j.ydbio.2019.10.011 (2019).
- 8 Laranjeira, C. *et al.* Glial cells in the mouse enteric nervous system can undergo neurogenesis in response to injury. *J Clin Invest* **121**, 3412-3424, doi:10.1172/JCI58200 (2011).
- 9 McCallum, S. *et al.* Enteric glia as a source of neural progenitors in adult zebrafish. *Elife* **9**, doi:10.7554/eLife.56086 (2020).
- 10 Bondurand, N., Natarajan, D., Thapar, N., Atkins, C. & Pachnis, V. Neuron and glia generating progenitors of the mammalian enteric nervous system isolated from foetal and postnatal gut cultures. *Development* **130**, 6387-6400, doi:10.1242/dev.00857 (2003).
- 11 Kruger, G. M. *et al.* Neural crest stem cells persist in the adult gut but undergo changes in self-renewal, neuronal subtype potential, and factor responsiveness. *Neuron* **35**, 657-669 (2002).
- 12 Durbec, P. L., Larsson-Blomberg, L. B., Schuchardt, A., Costantini, F. & Pachnis, V. Common origin and developmental dependence on c-ret of subsets of enteric and sympathetic neuroblasts. *Development* **122**, 349-358 (1996).

REVIEWERS' COMMENTS

Reviewer #1 (Remarks to the Author):

The reviewer thanks the authors for their thoughtful responses to the points raised in the initial review. The revised submission is a beautiful contribution and all of my concerns have been addressed. Congratulations on a very nice study.

Reviewer #2 (Remarks to the Author):

The authors have responded to this Reviewer's comments. The revised manuscript is improved and addresses prior concerns.

Reviewer #3 (Remarks to the Author):

My comments have been addressed, and I recommend this revised manuscript for publication.